# Network instability dynamics drive a transient bursting period in the developing hippocampus in vivo

Jürgen Graf[1†], Vahid Rahmati[1,2,3†], Myrtill Majoros[1], Otto W Witte[1], Christian Geis[1,2], Stefan J Kiebel[3], Knut Holthoff[1‡], Knut Kirmse[1,4*‡]

[1]Department of Neurology, Jena University Hospital, Jena, Germany; [2]Section Translational Neuroimmunology, Jena University Hospital, Jena, Germany; [3]Department of Psychology, Technical University Dresden, Dresden, Germany; [4]Department of Neurophysiology, Institute of Physiology, University of Würzburg, Würzburg, Germany

**Abstract** Spontaneous correlated activity is a universal hallmark of immature neural circuits. However, the cellular dynamics and intrinsic mechanisms underlying network burstiness in the intact developing brain are largely unknown. Here, we use two-photon $Ca^{2+}$ imaging to comprehensively map the developmental trajectories of spontaneous network activity in the hippocampal area CA1 of mice in vivo. We unexpectedly find that network burstiness peaks after the developmental emergence of effective synaptic inhibition in the second postnatal week. We demonstrate that the enhanced network burstiness reflects an increased functional coupling of individual neurons to local population activity. However, pairwise neuronal correlations are low, and network bursts (NBs) recruit CA1 pyramidal cells in a virtually random manner. Using a dynamic systems modeling approach, we reconcile these experimental findings and identify network bi-stability as a potential regime underlying network burstiness at this age. Our analyses reveal an important role of synaptic input characteristics and network instability dynamics for NB generation. Collectively, our data suggest a mechanism, whereby developing CA1 performs extensive input-discrimination learning prior to the onset of environmental exploration.

*For correspondence:
knut.kirmse@uni-wuerzburg.de

†These authors contributed equally to this work

‡Senior Author

**Competing interest:** The authors declare that no competing interests exist.

## Editor's evaluation

This study provides fundamental findings about the developing brain and compelling evidence for how hippocampal physiology evolves during the first few postnatal weeks. Unlike previous in vitro results, which find declining network synchrony after the first postnatal week, the authors find in vivo that synchrony increases and peaks in the second postnatal week, despite emerging GABA-mediated inhibition during this time. They develop a model to explain these findings and suggest an underlying bistable population dynamic, oscillating between silent and active states, that sculpts input discrimination and network synchrony.

## Introduction

Developing neural circuits generate correlated spontaneous activity in which co-activations of large groups of neurons are interspersed by relatively long periods of quiescence (*Kirmse and Zhang, 2022*; *Molnár et al., 2020*). In rodents, network activity commences long before the onset of hearing, vision, and active environmental exploration and makes important contributions to the proper assembly of brain circuits (*Kirkby et al., 2013*). Activity-dependent refinements operate at multiple steps of

maturation, including the control of neural progenitor progression (*Vitali et al., 2018*), apoptotic cell death (*Blanquie et al., 2017*; *Wong et al., 2018*), neuronal cell-type specification (*Sun et al., 2018*), migration (*Maset et al., 2021*) as well as synapse formation and plasticity (*Oh et al., 2016*; *Sando et al., 2017*; *Winnubst et al., 2015*). Experimental and theoretical evidence suggest that, in addition to the overall level of activity, specific spatiotemporal firing patterns are critical for activity-dependent refinements to occur (*Albert et al., 2008*; *Zhang et al., 2011*).

A representative example of correlated spontaneous network activity is found in the neonatal hippocampus in vivo. During the first postnatal week, the main electrophysiological signature is bursts of multi-unit activity (*Leinekugel et al., 2002*), which bilaterally synchronize large parts of the dorsal CA1 and are often accompanied by sharp waves (SPWs) in the local field potential (*Valeeva et al., 2019*; *Valeeva et al., 2020*). SPWs frequently follow myoclonic limb or whisker twitches (*Dard et al., 2022*; *Del Rio-Bermudez et al., 2020*; *Karlsson et al., 2006*; *Valeeva et al., 2019*), suggesting that SPWs convey feedback information from the somatosensory periphery. By the second postnatal week, discontinuous activity in the olfactory bulb drives network oscillations in the entorhinal cortex (*Gretenkord et al., 2019*), further pointing to a role of multi-sensory integration in limbic ontogenesis. In the neonatal CA1, recent in vivo investigations revealed that GABAergic interneurons (INs) could promote a second class of SPW-independent network events through NKCC1-dependent chloride uptake in pyramidal cells (PCs), although inhibitory effects of GABAergic signaling coexist (*Graf et al., 2021*; *Murata and Colonnese, 2020*; but see *Valeeva et al., 2016*). A qualitatively similar situation applies to the immature hippocampus in vitro (*Ben-Ari et al., 1989*; *Flossmann et al., 2019*), in which correlated spiking of PCs is facilitated by increasing the intracellular chloride concentration (*Spoljaric et al., 2019*; *Zhang et al., 2019*), whereas inhibition of chloride uptake has the opposite effect (*Dzhala et al., 2005*). In this line, in vitro studies suggest that correlated spontaneous activity largely disappears by the beginning of the second postnatal week, when the reversal potential of $GABA_A$ receptor-mediated currents shifts into the hyperpolarizing direction (*Spoljaric et al., 2017*; *Tyzio et al., 2008*). However, the developmental trajectories of cellular network firing dynamics in the hippocampus in vivo remain largely unknown.

Using two-photon $Ca^{2+}$ imaging, we here provide a detailed analysis of the spatiotemporal dynamics of network activity in the developing CA1 region at single-cellular resolution in vivo. We reveal that CA1 PCs undergo a transient period of enhanced burst-like network activity during the second postnatal week, when GABA already acts as an inhibitory transmitter. Our results show that, at this time, network bursts (NBs) recruit CA1 PCs in an almost random manner, and recurring cellular activation patterns become more stable only after eye opening. Using computational network modeling, we identify bi-stability as a dynamical regime underpinning the enhanced bursting activity of CA1 PCs. We show that NBs mainly reflect the network's intrinsic instability dynamics, which exquisitely depend on proper input timing and strength. In addition, inhibitory GABAergic signaling effectively promotes state transitions underlying NB generation. Our data suggest a mechanism, whereby CA1 undergoes extensive input-discrimination learning before the onset of environmental exploration.

## Results
### Reliable detection of somatic $Ca^{2+}$ transients in densely labeled tissue

We used in vivo two-photon laser-scanning microscopy (2PLSM) in spontaneously breathing, head-fixed $Emx1^{IREScre}$:$Rosa26^{LSL-GCaMP6s}$ mice to record somatic $Ca^{2+}$ transients (CaTs) from CA1 PCs as a proxy of their firing activities. In this strain, Cre is expressed in virtually all CA1 PCs (*Gorski et al., 2002*; *Kummer et al., 2012*). Due to the finite point-spread function inherent to 2PLSM, dense cell labeling resulted in a non-negligible overlap of signals originating from neighboring somata and/or neurites. Our preliminary analysis revealed that standard CaT detection methods based on analyzing mean fluorescence intensities from regions of interests (ROIs) can lead to substantial false positive rates (*Figure 1*). Likewise, recent studies have demonstrated that popular CaT analysis algorithms can produce substantial misattribution errors under such conditions (*Chen et al., 2020*; *Denis et al., 2020*; *Gauthier et al., 2022*). We therefore devised a novel cell-specific spatial template-matching approach for the reliable detection of CaTs in densely labeled tissue, which we refer to as CATHARSiS (*C*alcium *t*ransient detection *har*nessing *s*patial *si*milarity). CATHARSiS makes use of the fact that, for each cell, the spike-induced changes in GCaMP fluorescence intensity ($\Delta F$) have a spatially

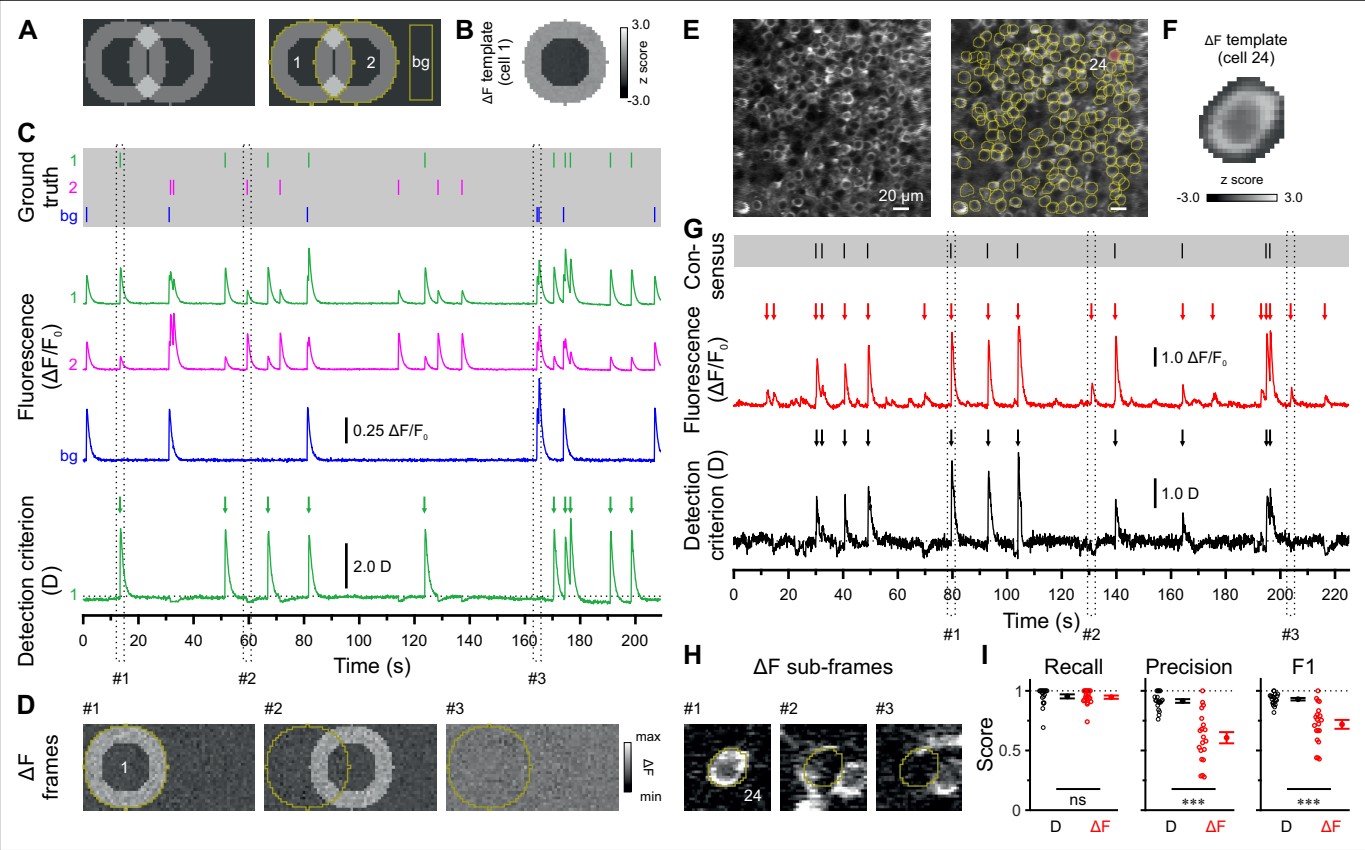

**Figure 1.** *Ca*lcium *t*ransient detection *har*nessing *s*patial *si*milarity (CATHARSiS) enables reliable Ca²⁺ transient (CaT) detection in densely labeled tissue. (**A**) Resting image of two partially overlapping simulated cells (*left*) and regions of interest (ROIs) used for analysis (*right*). bg – background. (**B**) ΔF template of cell 1. (**C**) Top, simulated trains of action potentials. Middle, relative changes from baseline fluorescence (ΔF/F₀) of ROIs shown in A. Bottom, detection criterion (D) for cell 1 and corresponding CaT onsets retrieved by CATHARSiS (arrows). (**D**) Sample ΔF images of three individual frames at time points indicated in C. Spikes in cells 1 or 2 translated into ring-shaped increases in ΔF, whereas those induced by bg spikes were applied to the entire field of view. (**E**) Resting GCaMP6s fluorescence image (*left*) and ROIs used for analysis (*right*). (**F**) ΔF template of the cell indicated in E. (**G**) Top, consensus visual annotation by two human experts for the same cell. Middle, ΔF/F₀ and detected event onsets (red arrows). Bottom, detection criterion (D) and corresponding CaT onsets retrieved by CATHARSiS (black arrows). (**H**) Sample ΔF images of three individual frames at time points indicated in G. Note that frames #2 and #3 led to false positive results if event detection was performed on mean ΔF, but not if performed on D. (**I**) Quantification of recall, precision, and F1 score for event detection based on D (i.e. CATHARSiS) and mean ΔF, respectively. Each open circle represents a single cell (n = 20 cells). Data are presented as mean ± SEM. ns – not significant. *** p<0.001. See also ***Supplementary file 1a*** and ***Figure 1—source data 1***.

The online version of this article includes the following source data for figure 1:

**Source data 1.** Numerical data presented in ***Figure 1***.

inhomogeneous (ring-like) configuration (see Methods for details). In brief, a cell-specific spatial ΔF template representing the active cell is computed (***Figure 1A and B***) and optimally scaled to fit its ΔF in each recorded frame. Based on the optimum scaling factor and the quality of the fit, a detection criterion $D(t)$ is computed for each time point (***Clements and Bekkers, 1997***). $D(t)$ is then subjected to a general-purpose event detection routine for the extraction of CaT onsets (***Rahmati et al., 2018***). We first illustrate CATHARSiS by analyzing simulated spike-induced CaTs in ring-shaped cells (***Figure 1A and B***). Here, fluorescence signals of the cell of interest were contaminated by: (1) signals originating from a partially overlapping second cell, (2) spatially homogenous fluorescence changes mimicking axon-based neuropil activity (***Kerr et al., 2005***), and (3) a low level of Poissonian noise (***Figure 1C***). ***Figure 1C and D*** demonstrate that $D(t)$ will increase only if ΔF has a spatial configuration similar to that of the template, i.e., if the simulated cell is active. Of note, $D(t)$ is insensitive to a spatially uniform offset of ΔF and can decrease for mean ΔF increases having a dissimilar spatial configuration (#2 and #3 in ***Figure 1C and D***). We applied CATHARSiS to two simulated sample cells of identical shape and varied their spatial overlap from 0 to 75% of the cell area, in accordance to the observed overlap in

our empirical data. CATHARSiS correctly retrieved all ground-truth CaTs without false positive events (n=665 CaTs in total). We also found that the delay of the detected CaT onsets vs. simulated spikes was low (–0.3±0.0 frames), pointing to a high temporal accuracy of spike reconstruction, which is a prerequisite of a precise analysis of spatiotemporal activity patterns. We next evaluated CATHARSiS on data recorded from developing CA1 PCs in $Emx1^{IREScre}$:$Rosa26^{LSL-GCaMP6s}$ mice in vivo (*Figure 1E–I*). For comparison, a consensus visual annotation by human experts was used (*Figure 1G*, top), as simultaneous electrophysiological data were not available (see Methods). We compared CATHARSiS to an event detection routine based on analyzing mean ΔF(t) and found that recall was ~95% for both approaches (*Figure 1I*; *Supplementary file 1a*). However, CATHARSiS yielded considerably fewer false positive events, thus resulting in a significantly higher precision and F1 score (*Figure 1I*). Importantly, the delay of detected CaT onsets relative to the consensus annotation was consistently low (0.9±0.1 frames at a frame rate of 11.6 Hz, n=20 cells), confirming that CATHARSiS achieved a high temporal accuracy.

We conclude that CATHARSiS is suited for the reliable reconstruction of somatic CaTs in densely packed neuronal tissue with both high detection and temporal accuracies.

## A transient period of firing equalization during CA1 development in vivo

In the adult CA1, firing rate distributions are approximately log-normal, implying that a minority of neurons accounts for the majority of spikes. In addition, firing rates of individual cells are relatively stable across brain states and tasks, suggesting that skewed firing rate distributions reflect an inherent characteristic of mature hippocampal computations (*Mizuseki and Buzsáki, 2013*). To reveal developmental trajectories of single-cell activity, we applied CATHARSiS to extract spontaneous CaTs from *Emx1+* PCs at P3–4 (n=19 fields of view [FOVs] from six mice), P10–12 (n=11 FOVs from six mice), and P17–19 (n=12 FOVs from six mice; *Figure 2A*). For the sake of brevity, these age groups are hereafter referred to as P4, P11, and P18, respectively. We found that mean CaT frequencies significantly increased ~2.5-fold from 1.5±0.2 min$^{-1}$ at P4 to 3.9±0.4 min$^{-1}$ at P11 and remained relatively stable afterward (P18: 4.8±0.3 min$^{-1}$, *Figure 2B and C*; see *Supplementary file 1b*). Additionally, we observed a striking change in the shape of CaT frequency distributions, which were broad and strongly right-tailed at P4 and P18, but much less so in the second postnatal week (*Figure 2B*; #6 in *Supplementary file 1b*). To quantify the dispersion of CaT frequencies among individual cells, we plotted the corresponding Lorenz curves (*Figure 2D*), in which the cumulative proportion of CaT frequencies is plotted against the cumulative proportion of cells rank-ordered by frequency (*Mizuseki and Buzsáki, 2013*). Here, the line of equality represents the case where all neurons have equal CaT frequencies. We computed the Gini coefficient as a measure of deviation from equality (for a graphical representation, see inset in *Figure 2D*). Gini coefficients underwent a transient minimum at P11, indicating that CaT frequencies among individual neurons were considerably more similar to each other as compared to P4 and P18 (*Figure 2E*). We next addressed whether developmental alterations in average CaT frequencies were accompanied by changes in their irregularity of occurrence. The local coefficient of variation (CV2), a robust measure of local spiking irregularity (*Holt et al., 1996*; *Ponce-Alvarez et al., 2010*), gradually declined from P4 to P18 (*Figure 2G*). At P11, CV2 was close to one, indicating that the irregularity of CaT occurrence is similar to that of a Poissonian point process, in which successive events occur independently of one another. As previously observed for CaT frequencies, CV2 distributions were also relatively broad at P4 and P18, but narrow at P11 (*Figure 2F*). Consistently, Gini coefficients of CV2 showed a distinct minimum at P11 (see #4 in *Supplementary file 1b*).

Collectively, our data reveal a transient equalization in CaT statistics of individual CA1 PCs during the second postnatal week, while highly skewed CaT frequency distributions eventually emerge only around/after eye opening.

## CA1 undergoes a transient enhanced bursting period while progressively transitioning from discontinuous to continuous activity

Previous in vitro work has identified giant depolarizing potentials (GDPs) as the most prominent pattern of correlated network activity in the neonatal hippocampus (*Ben-Ari et al., 1989*; *Garaschuk et al., 1998*; *Leinekugel et al., 1997*). GDPs depend on a depolarizing action of GABA$_A$ receptor-dependent transmission (*Ben-Ari et al., 1989*; *Owens et al., 1996*) and disappear at around the

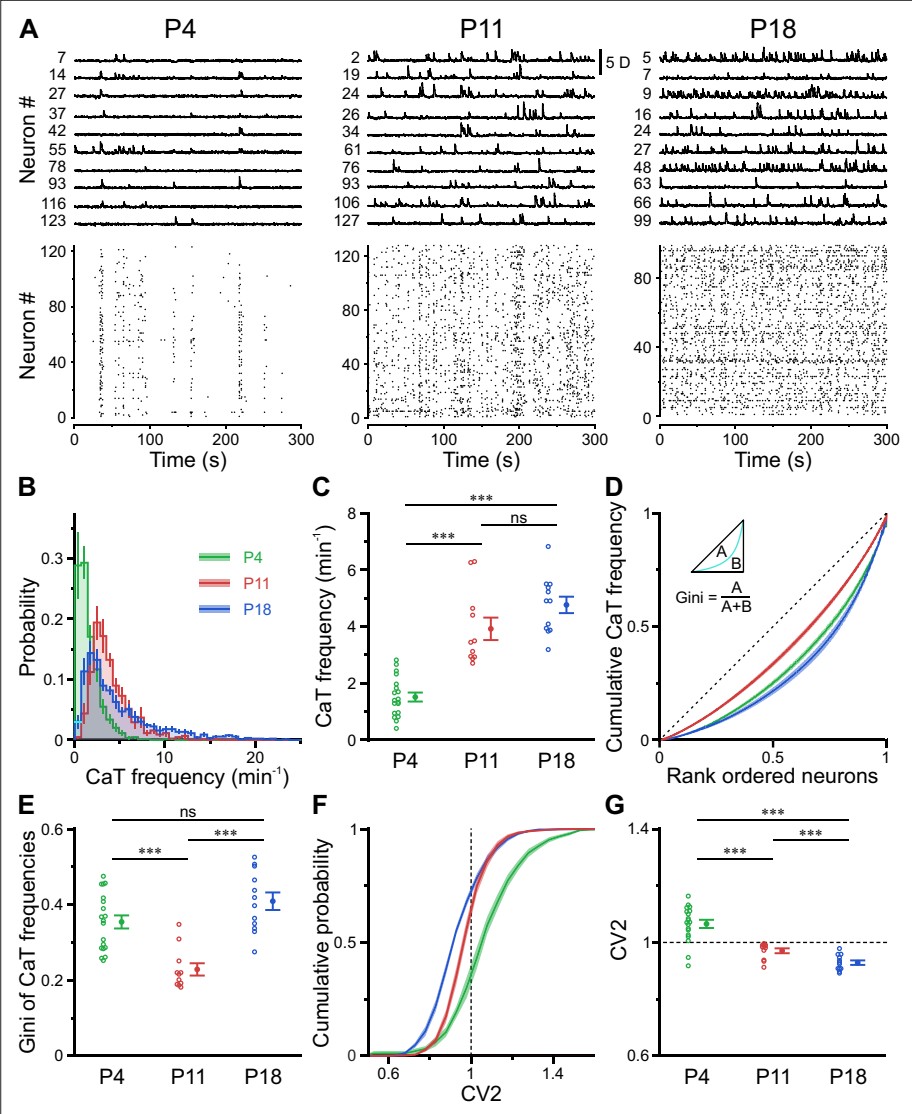

**Figure 2.** A transient period of firing equalization during CA1 development in vivo. (**A**) Sample *D(t)* traces (top) and raster plots showing reconstructed Ca²⁺ transient (CaT) onsets (bottom). Note the developmental transition from discontinuous to continuous network activity. (**B**) Empirical probability distribution of CaT frequencies. (**C**) Mean CaT frequencies per field of view (FOV). (**D**) Lorenz curves of CaT frequencies. Line of equality (dotted) represents the case that all neurons have equal CaT frequencies. Inset depicts Gini coefficient calculation. (**E**) Mean Gini coefficients per FOV. (**F**) Cumulative probability of mean coefficient of variation (CV2) of inter-CaT intervals. Note that, at P11, CV2 distribution is narrower and centered around 1. (**G**) Mean CV2 per FOV. For a Poisson process, CV2 = 1 (dotted line). Each open circle represents a single FOV. Data are presented as mean ± SEM. ns – not significant. P4: P3–4, n = 19 FOVs from six mice, P11: P10–12, n = 11 FOVs from six mice, P18: P17–19, n = 12 FOVs from six mice, and *** p<0.001. See also *Supplementary file 1b* and *Figure 2—source data 1*.

The online version of this article includes the following source data for figure 2:

**Source data 1.** Numerical data presented in *Figure 2*.

beginning of the second postnatal week, when GABA actions shift from mainly excitatory to mainly inhibitory (*Tyzio et al., 2007*; *Yamada et al., 2004*). To investigate whether a similar developmental profile of NB generation exists in the CA1 in vivo, we next determined the time-course of the fraction of active cells Φ(t) (*Figure 3A*). At P4, CA1 PCs spent relatively long time periods in a low-activity (silent) state, which was only interspersed by brief NBs (*Figure 3A*, left, and *Figure 3B*). During NBs, Φ(t) rarely exceeded 20% (*Figure 3B*), indicating that the degree of co-activation in vivo is considerably lower than that reported for GDPs in vitro (*Flossmann et al., 2019*; *Garaschuk et al., 1998*;

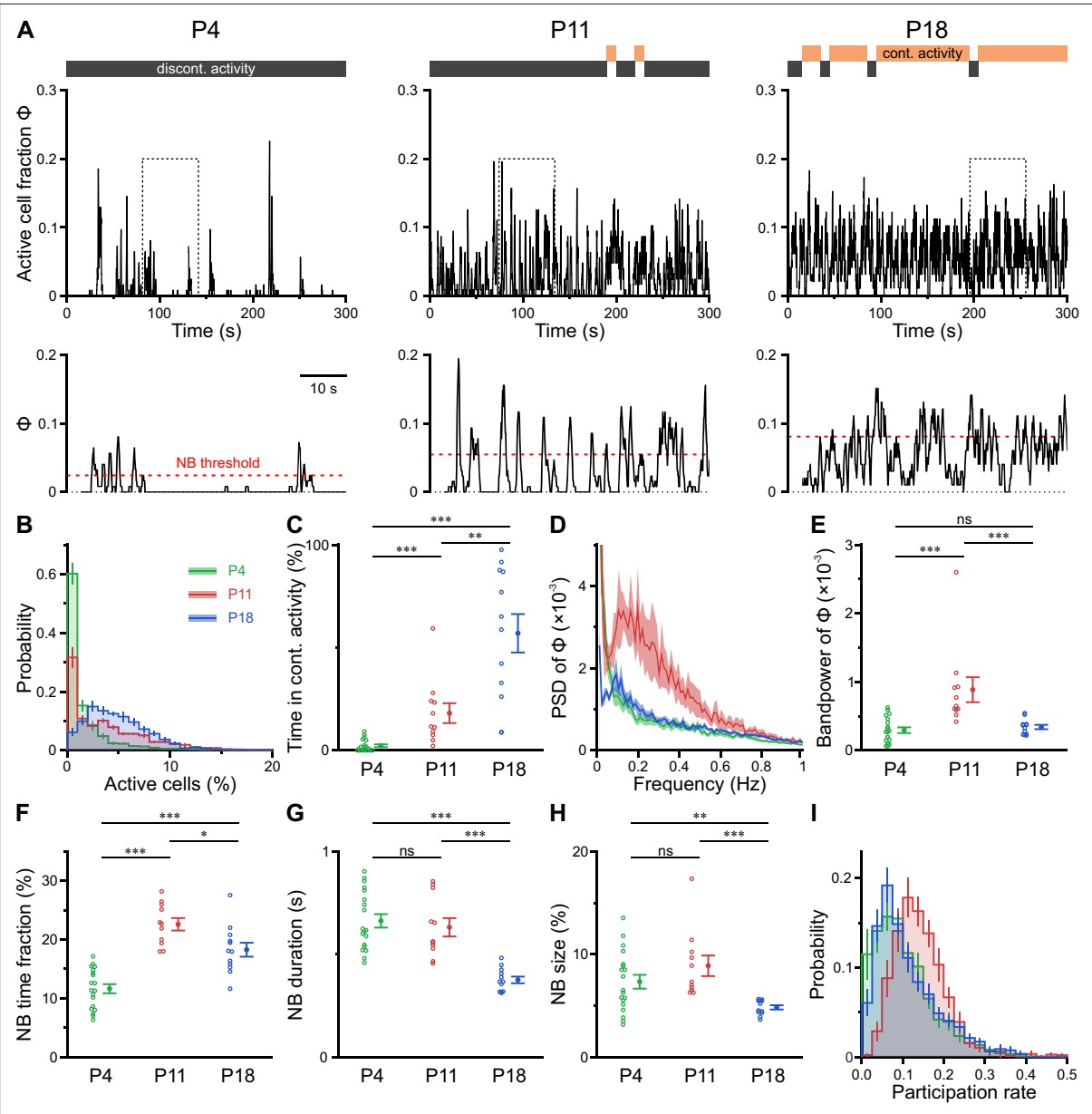

**Figure 3.** CA1 undergoes a transient enhanced bursting period while progressively transitioning from discontinuous to continuous activity. (**A**) Sample traces of the fraction of active cells Φ(t). Gray and orange bars indicate discontinuous and continuous network activity, respectively. Bottom traces show time periods marked on top (dotted rectangle) at higher temporal resolution. Red dotted lines indicate the activity-dependent thresholds for network burst (NB) detection. (**B**) Empirical probability distribution of active cells per frame. (**C**) Continuous activity emerges at P11 and dominates over discontinuous activity at P18. (**D**) Power spectral density of Φ(t). (**E**) Bandpower of Φ(t) in the 0.1–0.5 Hz range. (**F**) The fraction of time that the network spends in NBs peaks at P11. (**G**) The average NB duration is lowest at P18. (**H**) Quantification of NB size as the mean fraction of active neurons (corrected for NB threshold as indicated in A). (**I**) Empirical probability distribution of the fraction of NBs that each cell is participating in. Each open circle represents a single field of view (FOV). Data are presented as mean ± SEM. ns – not significant. P4: P3–4, P11: P10–12, P18: P17–19, *** p<0.001, ** p<0.01, and * p<0.05. See also ***Supplementary file 1c*** and ***Figure 3—source data 1***.

The online version of this article includes the following source data and figure supplement(s) for figure 3:

**Source data 1.** Numerical data presented in ***Figure 3***.

**Figure supplement 1.** Developmental changes in network burst (NB) characteristics are robust to a wide range of definitions.

**Figure supplement 1—source data 1.** Numerical data presented in ***Figure 3—figure supplement 1***.

*Leinekugel et al., 1997*). In contrast to GDPs, bursting activity in vivo was even more pronounced at P11, when the network tended to oscillate between a silent state and a bursting mode with an inter-burst period of ~2–10 s (*Figure 3A*, middle). We next asked if CA1 PCs could maintain non-zero activity for longer time periods. To this end, we partitioned recordings into non-overlapping 10-s-long windows, which we classified as either continuous or discontinuous based on a threshold criterion applied to $\Phi(t)$ (see Methods for details). Whereas network activity was found to be entirely discontinuous in the first postnatal week, CA1 started to dynamically transition between discontinuous and continuous activity at P11 (*Figure 3A and C*). At this age, the proportion of continuous activity was, on average, low but substantially increased toward P18 (*Figure 3C* and *Supplementary file 1c*).

To quantify developmental changes in the rhythmicity of network activity, we first computed the power spectrum of $\Phi(t)$. At P11, this revealed a distinct peak in the range of ~0.1–0.5 Hz (*Figure 3D*), pointing to the existence of a preferred oscillation frequency of CA1 PCs. Such a power peak was absent at P4 and reduced at P18. Accordingly, band power in the 0.1–0.5 Hz frequency range was significantly higher in the second postnatal week than at earlier or later stages (*Figure 3E*).

To characterize periods of neuronal co-activation in more detail, we next extracted NBs by thresholding $\Phi(t)$. NB thresholds were determined separately for each FOV using surrogate data, accounting for differences in CaT frequencies (*Figure 3A*, bottom, and Methods). The fraction of time that the network spent in NBs was lowest at P4 and peaked at P11 (*Figure 3F*). Moreover, NBs at P18 were significantly shorter in duration than during the first and second postnatal weeks (*Figure 3G*). We quantified NB size as the fraction of active cells (corrected for the threshold applied to $\Phi[t]$) and found that it only declined after P11 (*Figure 3H*). At P11, each neuron participated in 14.4 ± 1.2% of all NBs, which significantly exceeded participation rates at P4 (10.3 ± 0.8%) and P18 (11.2 ± 0.4%; see #6 in *Supplementary file 1c*). Importantly, these developmental changes in NB characteristics were robust to a wide range of NB definitions (*Figure 3—figure supplement 1* and Methods). Additionally, distributions of participation rates were relatively narrow at P11 (*Figure 3I*), pointing to a greater similarity of cells with respect to their contribution to NB generation as compared to earlier or later developmental stages (#7 in *Supplementary file 1c*).

Taken together, our data reveal that CA1 undergoes a transient period of enhanced bursting activity that coincides with the developmental appearance of continuous activity states during the second postnatal week. At this age, network activity displays rhythmicity in the sub-Hz range – in spite of the close-to-random activation of individual PCs.

## Enhanced population coupling underlies network burstiness in the second postnatal week in vivo

The transient developmental increase in bursting propensity was unexpected, as (1) GDPs in vitro disappear soon after the first postnatal week (*Ben-Ari et al., 1989*; *Garaschuk et al., 1998*; *Khazipov et al., 2004*) and (2) previous in vivo data from neocortex revealed a reduction in correlated network activity during the same time period (*Colonnese et al., 2010*; *Golshani et al., 2009*; *Rochefort et al., 2009*; *van der Bourg et al., 2017*). We therefore assessed whether the enhanced burstiness at P11 reflects an increase in functional neuronal coupling. We first investigated the coupling of single-cell activity to that of the overall population. For each cell, we computed its population coupling (PopC) index (*Okun et al., 2015*; *Sweeney and Clopath, 2020*) and tested for its significance using surrogate data (see Methods). The PopC index significantly peaked at P11 (*Figure 4A*, *Supplementary file 1d*), while there was no difference between P4 and P18. The higher PopC index at P11 arose from a significantly higher fraction of coupled cells (*Figure 4B*), whereas the indices of significantly coupled cells were similar (*Figure 4C*). We next addressed whether the increased PopC index at P11 results from an increase in pairwise temporal correlation of CaTs. To this end, we computed the spike-time tiling coefficient (STTC) as a frequency-independent affinity metric of two event time series (*Cutts and Eglen, 2014*; *Figure 4D*). The fraction of significantly correlated cell pairs did not significantly differ between P4 and P11 (P4: 13 ± 2% and P11: 23 ± 5%), but strongly decreased to 5 ± 1% at P18 (*Figure 4E*). STTCs of significantly correlated pairs profoundly declined from 0.18±0.01 at P4 to 0.09±0.01 already at P11, but did not significantly change afterward (P18: 0.10±0.00; *Figure 4F and G*). These data suggest that developmental changes in pairwise neuronal correlations do not account for the increased PopC or the increased burstiness of CA1 PCs during the second postnatal week. We further analyzed the spatial structure of CA1 ensemble dynamics. We found that the dependence of

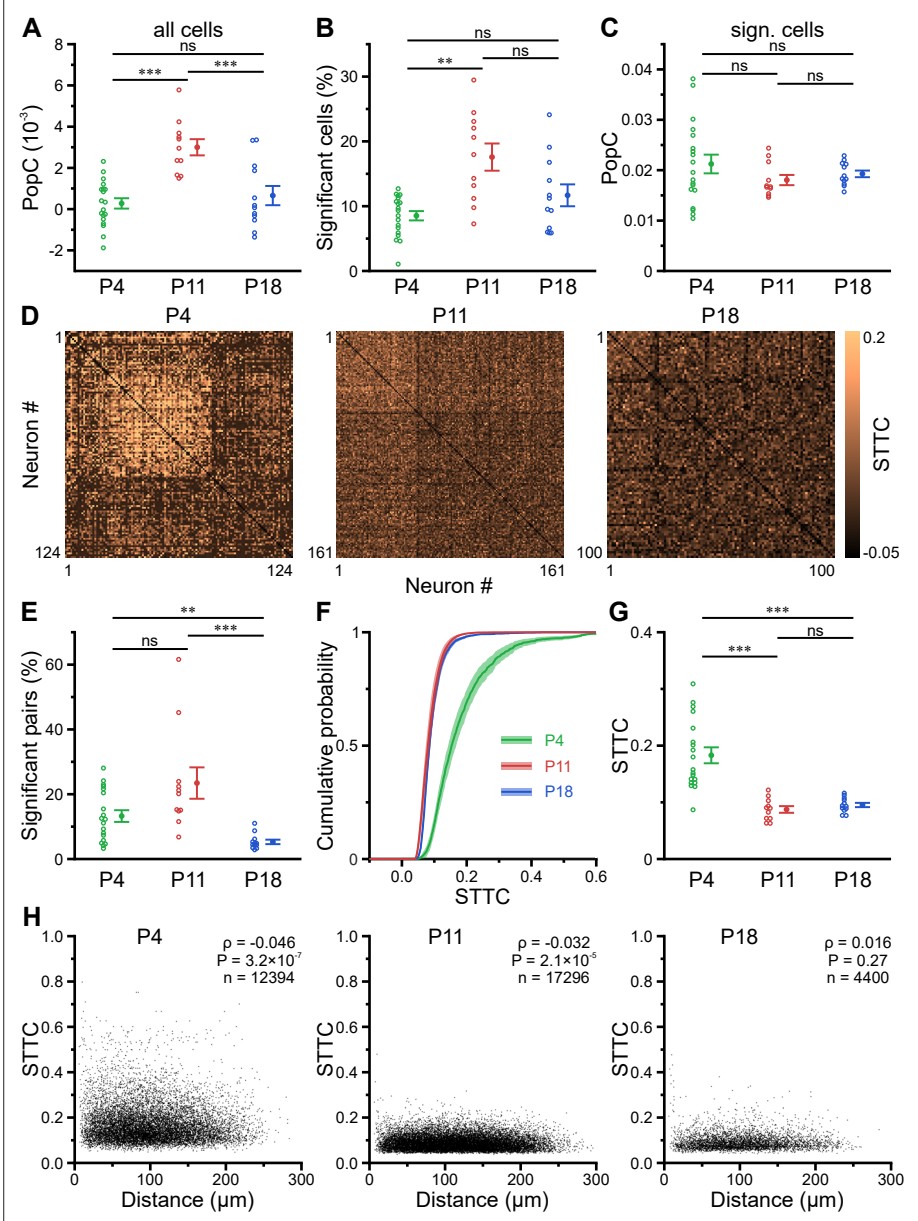

**Figure 4.** Enhanced population coupling (PopC) underlies network burstiness in the second postnatal week in vivo. (**A**) The mean PopC index peaked at P11. (**B**) Mean fraction of cells with significant PopC. (**C**) Mean PopC index of significantly coupled cells only. (**D**) Sample spike-time tiling coefficient (STTC) matrices (re-ordered) from three individual fields of view (FOVs) at P4, P11, and P18. (**E**) Mean fraction of cell pairs having a significant STTC. (**F**) Cumulative probability of STTCs of significantly correlated cell pairs only. (**G**) Mean STTCs of significantly correlated cell pairs. (**H**) Relationship between STTC and Euclidean somatic distance for significantly correlated cell pairs. $\rho$ denotes the Spearman's rank correlation coefficient for all cell pairs analyzed (**n**) at a given age. (**A–C** and **E–G**) Each open circle represents a single FOV. Data are presented as mean ± SEM. ns – not significant. P4: P3–4, P11: P10–12, P18: P17–19, *** $p<0.001$, and ** $p<0.01$. See also ***Supplementary file 1d*** and ***Figure 4—source data 1***.

The online version of this article includes the following source data for figure 4:

**Source data 1.** Numerical data presented in ***Figure 4***.

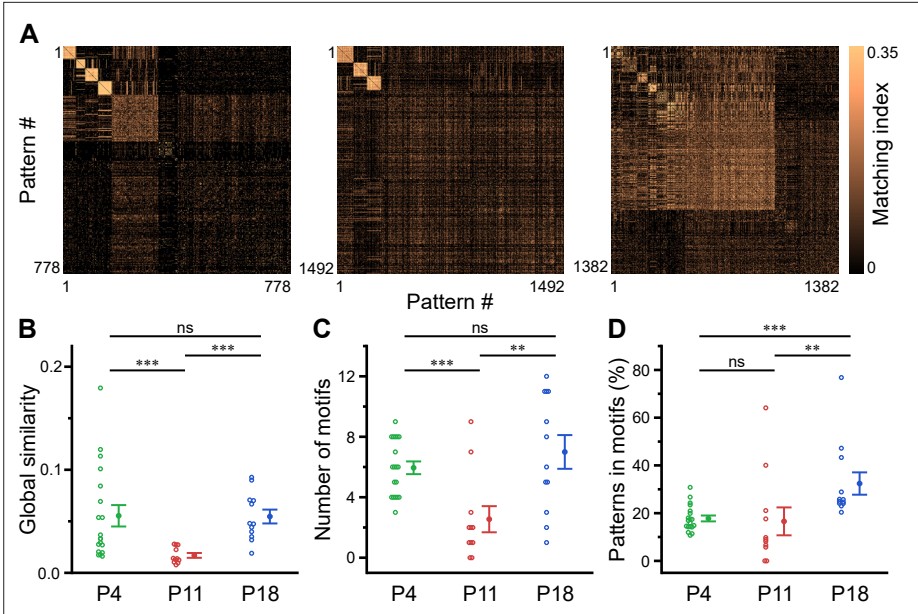

**Figure 5.** Motifs of CA1 network activity undergo distinct developmental alterations. (**A**) Similarity matrices (matching index) of binary activity patterns (re-ordered for illustration of motif detection) from three individual fields of view (FOVs) at P4, P11, and P18. (**B**) Global similarity of activity patterns is lowest at P11. (**C**) The absolute number of detected motifs per FOV is lowest at P11. (**D**) Motif repetition quantified as the fraction of activity patterns belonging to each motif. Each open circle represents a single FOV. Data are presented as mean ± SEM. ns – not significant. P4: P3–4, P11: P10–12, P18: P17–19, *** p<0.001, and ** p<0.01. See also *Supplementary file 1e* and *Figure 5—source data 1*.

The online version of this article includes the following source data for figure 5:

**Source data 1.** Numerical data presented in *Figure 5*.

STTCs on the Euclidean somatic distance was weak already during the first two postnatal weeks and non-significant at P18 (*Figure 4H*), indicating that the horizontal confinement of patterned network activity is weak or absent throughout the developmental period studied here.

Collectively, our data reveal that enhanced network burstiness during the second postnatal week is associated with a higher fraction of cells significantly locked to the activity of the local network, while pairwise neuronal correlations are low.

## Motifs of CA1 network activity undergo distinct developmental alterations

Recurring spatiotemporal cellular activation patterns are a hallmark of network activity in the adult hippocampus in vivo (*Villette et al., 2015*). Whether such repeating patterns (hereafter referred to as 'motifs') are already present at early developmental stages is unknown. To detect motifs, we divided the recording time into non-overlapping bins, each represented by a binary spatial pattern (vector) of active and inactive cells, followed by computing the matching index matrix of all possible pattern pairs (*Figure 5A*). We then applied an eigendecomposition-based clustering method to each similarity matrix in order to detect potential motifs, while testing for their significance using surrogate data (see Methods). First, this analysis revealed that the global similarity of the activation patterns was lowest at P11 (*Figure 5B*, *Supplementary file 1e*), whereas it was similar between P4 and P18. This finding implies that there is less commonality between the sets of active cells present in different patterns at P11 and, thus, more random recruitment of cells. Furthermore, the number of motifs was significantly lower at P11 (2.5±0.9) as compared to P4 (5.9±0.4) and P18 (7.0±1.1; *Figure 5C*). When computing the fraction of patterns belonging to each motif, we found that the motifs had the highest repetition rate at P18 (32.0 ± 4.7%), while there was no significant difference between P4 (17.8 ± 1.2%) and P11 (16.6 ± 5.9%; *Figure 5D*). The increase in motif repetition rate toward P18 suggests that recurring cellular activation patterns become more stable only after the onset of environmental exploration.

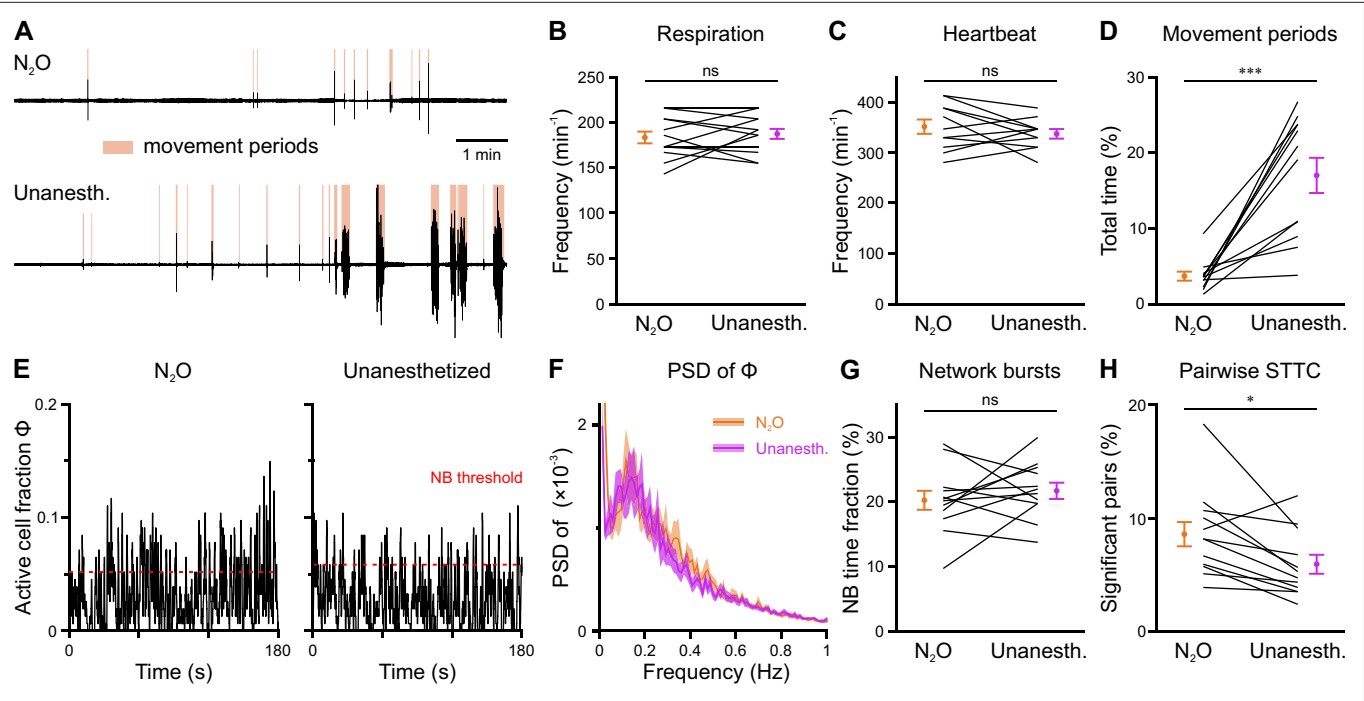

**Figure 6.** Effects of nitrous oxide ($N_2O$) on body movements, vital parameters, and network activity at P11. (**A**) Sample respiration/movement signal from an individual mouse receiving either 75% $N_2O$/25% $O_2$ (top) or pure $O_2$ (Unanesth., bottom). Detected movement periods are highlighted. (**B**) Respiration rate is unaffected by $N_2O$ (n = 12 fields of view [FOVs] from six mice). (**C**) Heart rate is unaffected by $N_2O$ (n = 11 FOVs; in the recording from one FOV, heart rate could not be reliably determined). (**D**) $N_2O$ significantly reduces movement periods. (**E**) Sample traces of the fraction of active cells $\Phi(t)$ ($\Delta t$ = 3 frames). Red dotted lines indicate the activity-dependent thresholds for network burst (NB) detection. (**F**) Power spectral density of $\Phi(t)$ is similar in the presence or absence of $N_2O$. (**G**) The fraction of time that the network spends in NBs. (**H**) Fraction of neuron pairs having a significant spike-time tiling coefficient (STTC). Data are presented as mean ± SEM. ns – not significant. *** p<0.001 and * p<0.05. See also *Supplementary file 1f* and *Figure 6—source data 1*.

The online version of this article includes the following source data and figure supplement(s) for figure 6:

**Source data 1.** Numerical data presented in *Figure 6* and *Figure 6—figure supplement 2*.

**Figure supplement 1.** Detection of body movements, breathing, and heart rate.

**Figure supplement 2.** Effects of nitrous oxide on CA1 network dynamics at P11.

## Effects of nitrous oxide on body movements, vital parameters, and network dynamics

In an attempt to reduce spontaneous movements and thus increase mechanical stability during two-photon imaging, all measurements discussed so far were performed in the presence of the analgesic-sedative nitrous oxide, following our established procedures in neonatal mice (*Kirmse et al., 2015*). However, general anesthetics can have a profound impact on brain activity in an age- and dose-dependent manner (*Ackman and Crair, 2014*; *Chini et al., 2019*; *Cirelli and Tononi, 2015*; *Yang et al., 2021*). As the effects of nitrous oxide on neuronal dynamics in the developing CA1 had been unknown, we performed a separate set of experiments (n=12 FOVs from six mice) in which we compared nitrous oxide to unanesthetized conditions using a paired design at P11 (*Figure 6—figure supplement 1*). Nitrous oxide reduced the time that mice spent in locomotion periods by several-fold (*Figure 6A and D* and *Figure 6—figure supplement 1*, *Supplementary file 1f*) and, hence, minimized periods of z-drift that needed to be discarded from analysis in two-photon $Ca^{2+}$ imaging. Unlike most conventional anesthetics, however, nitrous oxide did not affect respiration or heart rate (*Figure 6B and C* and *Figure 6—figure supplement 1*). To prevent photo-bleaching in these longer-lasting experiments, we minimized the laser power and increased the detector gain, which effectively reduced the signal-to-noise ratio in CaT detection. We observed slightly higher mean CaT frequencies in unanesthetized mice (*Figure 6E*, *Figure 6—figure supplement 2B and C*), while the Gini coefficient of CaT frequencies was unaffected (*Figure 6—figure supplement 2D*). The time the network

spent in continuous activity was similarly low (*Figure 6—figure supplement 2E*), and the oscillatory dynamics of CaTs (*Figure 6F* and *Figure 6—figure supplement 2F*) and NB properties (*Figure 6G* and *Figure 6—figure supplement 2G*) were unaltered. The most consistent effect of nitrous oxide was an increase in the percentage of correlated neuron pairs (*Figure 6H*), whereas their STTC values were unaffected (*Figure 6—figure supplement 2H*). This further translated into a higher fraction of cells that are significantly locked to local CA1 activity under nitrous oxide as compared to unanesthetized conditions (*Figure 6—figure supplement 2I*).

In sum, nitrous oxide moderately reduces pairwise and population coupling of CA1 PCs, whereas network burstiness and continuity resemble those observed in unanesthetized mice. A major advantage of nitrous oxide lies in the reduction of extended movement periods, which significantly increases mechanical stability during two-photon imaging.

## A neural network model with inhibitory GABA identifies intrinsic instability dynamics as a key to the emergence of NBs

Hitherto our analyses of experimental data revealed an unexpected bursting behavior of CA1 PCs at P11, despite the developmental emergence of synaptic inhibition (*Murata and Colonnese, 2020*; *Spoljaric et al., 2017*; *Tyzio et al., 2007*), which we related to their higher coupling to local network activity. However, the mechanisms governing in vivo network burstiness as well as its functional implications remain to be understood. Here, we provide mechanistic insights into these open questions by using computational network modeling and stability analysis techniques.

We employed a recurrent neural network (RNN) model of mean firing-activity rates of excitatory glutamatergic (PC) and inhibitory GABAergic (IN) cell populations ($A_{PC}$ and $A_{IN}$) with dynamic synaptic weights (*Flossmann et al., 2019*; *Rahmati et al., 2017*; *Figure 7A*). Here, we constrain the model with previously reported and our present experimental data obtained for P11: (1) GABAergic synapses are considered to be inhibitory (*Kirmse et al., 2015*; *Murata and Colonnese, 2020*; *Valeeva et al., 2016*) and (2) the spontaneous time-averaged $A_{PC}$ is effectively non-zero (*Figure 2C*). We found that such a network operates under a bi-stable regime, where two stable spontaneous fixed points (FPs) exist: one at a silent state ($A_{PC} = A_{IN} = 0$ Hz) and the other at an active state ($A_{PC} \neq 0$ Hz and $A_{IN} \neq 0$ Hz; green dots in *Figure 7B*). This is reminiscent of our experimental observations demonstrating that CA1 dynamically transitions between discontinuous and continuous activity states at this stage (see *Figure 3A–C*). The ability of the network model to embed the FP at an active state is mainly due to the stabilization function of inhibitory GABA (*Latham and Nirenberg, 2004*; *Rahmati et al., 2017*). Strikingly, our simulations showed that the network can process a given input quite differently at the silent and active states, respectively (time points a and c in *Figure 7C*). To this end, we applied a set of two excitatory inputs to the network's PC and IN populations ($e_{PC}$ and $e_{IN}$), resembling, e.g., SPW-driven inputs to CA1 (*Figure 7C*). We set the input strengths and duration to be identical across the two states. We found that, when operating at the active state, the network activity monotonically decays back to this state, once the input ceases (a in *Figure 7C*). However, at the silent state, input removal is followed by a transient profound surge in network activity (c in *Figure 7C*). Hereafter, we refer to this supra-amplification activity as simulated NB (simNB), emulating experimentally observed NBs (*Figure 3*).

## Network state-dependency of simNB generation

To disclose the mechanisms underlying this distinct behavior of the network at the silent and active state (*Figure 7C*), we computed the corresponding steady-state $A_{IN}$-$A_{PC}$-plane of the network, after freezing the slow short-term synaptic plasticity (STP) dynamics and, thus, synaptic weights (frozen STP-RNNs), at either of these states separately (*Figure 7D*). This analysis enables assessing the initial phase of network activity following an input perturbation. We found that, while operating at the silent state, the active state is not initially accessible to the network (lower panel in *Figure 7D*). Instead, an unstable FP is present in the network's fast (i.e. firing activity) dynamics, which builds an amplification threshold around the attraction domain of the FP located at the silent state. This in turn allows for the emergence of simNBs. A sufficiently strong perturbation, amenable to initially push the network activity beyond this threshold (i.e. to the amplification domain), will transiently expose the network to its intrinsic instability-driven dynamics, thereby effectively triggering a simNB (c in *Figure 7C*). Note that this unstable FP is different from its counterpart in the full system (*Figure 7B*) and is only visible

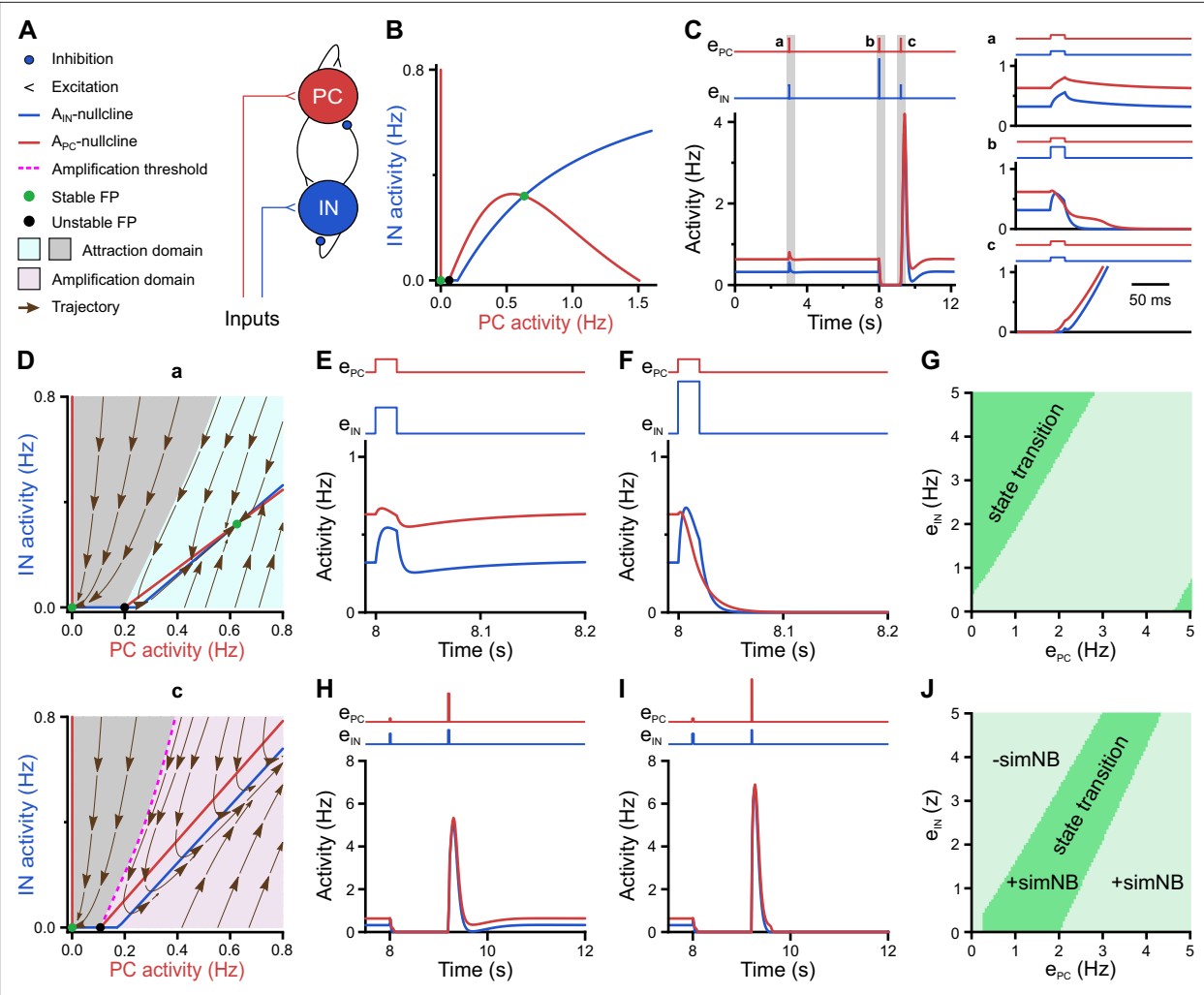

**Figure 7.** A neural network model with inhibitory GABA identifies intrinsic instability dynamics as key to the emergence of network bursts. (**A**) Schematic diagram of the short-term synaptic plasticity (STP)-recurrent neural network (RNN) model. (**B**) The $A_{IN}$-$A_{PC}$-plane of the full STP-RNN's stationary dynamics. Note the presence of two stable fixed points (FPs; green dots) at silent and active states as well as the unstable FP (black dot). (**C**) Simulated network burst (simNB) generation requires network silencing. The model was stimulated by pulse-like input to both pyramidal cell (PC) and interneuron (IN) populations for a duration of 0.020 s (at t = 3 and 9.2 s: $e_{PC} = e_{IN} = 0.25$; at t = 8 s:,$e_{PC} = 0.25$ $e_{IN} = 0.75$). Zoom-in of the activity around the stimulation times at active (a and b) and silent (c) states is shown in right panels. Input time series are shown on top of the plots. (**D**) The presence of an amplification domain in the initial phase of network firing dynamics enables the emergence of simNBs. The $A_{IN}$-$A_{PC}$-plane of the STP-RNN with synaptic efficacies frozen at active (a, top) and silent (b, bottom) states, right before input arrival. (**E–G**) Transition from active to silent state requires specific input ratios. Input delivered at t = 8 s. (**E**) A failed transition: $e_{PC} = 0.25$, $e_{IN} = 0.5$ (**F**) A successful transition: $e_{PC} = 0.25$, $e_{IN} = 1$. (**G**) A color-coded matrix of successful (dark green) and failed (light green) transitions to the silent state in response to different combinations of $e_{PC}$ and $e_{IN}$ amplitudes. (**H–J**) Both the transition from the silent to the active state and the simNB generation require specific input ratios. Input delivered at t = 9.2 s. Same as **E–G**, but for the backward transition to the active state. +simNB and –simNB indicate the emergence and absence of bursts.

in the network's fast dynamics. In particular, this FP is transient and disappears around the peak of the elicited simNB, mainly due to short-term synaptic depression (*Rahmati et al., 2017*). Unlike the silent state, the network frozen at the active state has no amplification domain, but instead two attraction domains related to its FPs at silent and active states (upper panel in *Figure 7D*). This explains the network's incapability of eliciting simNBs when operating at the active state. Collectively, these results suggest that simNBs, initiated by the input, are mainly an expression of the network's intrinsic instability dynamics, where the silent periods of the network are a prerequisite for its emergence. The model behavior agrees with our experimental data revealing both prominent burstiness and a dominance of discontinuous activity at P11 (see *Figure 3A–C*).

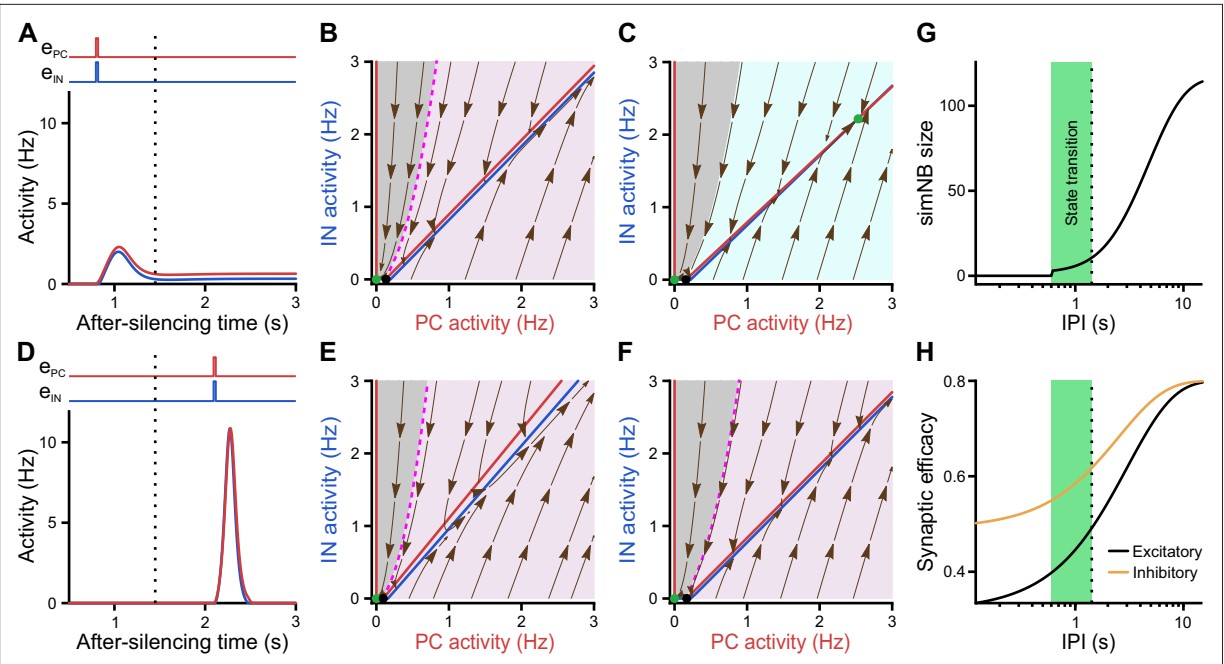

**Figure 8.** Internal deadline of state transitions. (**A–C**) Input delivered to the network before the deadline can move it to active state. (**A**) A successful transition. The input delivered at t = 0.8 s; $e_{PC} = 0.25$, $e_{IN} = 0.25$. (**B**) The $A_{IN}$-$A_{PC}$-plane of the short-term synaptic plasticity (STP)-recurrent neural network (RNN) with synaptic efficacies frozen at the silent state right before the input arrival. (**C**) Same as **B**, but frozen at the peak of the network burst (i.e. simulated network burst [simNB]) shown in A. Note the presence of the transient stable fixed point (FP; non-origin green dot), which triggers the transitioning to the active state. (**D–F**) Once the deadline is missed, the network cannot be moved to the active state by the subsequent input. Same as **A–C**, but the input delivered at t = 2.1 s. Note the absence of a non-origin transient stable FP in F, in contrast to C. (**G**) The simNB size and network transition to the active state depend on the inter-pulse intervals (IPIs: the arrival time of the next input relative to the silencing time of the network). simNB size is computed as the maximum of $A_{IN} + A_{PC}$ after the secondary input. Note the presence of a short window for transitioning to the active state. $e_{PC} = 0.25$, $e_{IN} = 0.25$. (**H**) Same as **G**, but for the non-scaled efficacies of GABAergic ($u_I x_I$; orange; see Methods) and glutamatergic ($u_P x_P$; black) synapses, right before the arrival of the secondary input. (**A, D, G, and H**) The dotted line at t = 1.45 s depicts the internal deadline.

The online version of this article includes the following figure supplement(s) for figure 8:

**Figure supplement 1.** Re-emergence of an internal deadline after a transition failure.

## Input-strength dependency and internal deadline of state transitions

What are the input requirements that allow the network to transition between the active and the silent states? First, we found that silencing the network from an active state requires specific ratios of excitatory input strengths to be delivered to its PC and IN populations (*Figure 7E–G*). In particular, the presence of GABAergic inhibition can effectively promote this transition, where otherwise a relatively much stronger $e_{PC}$ is required to silence the network solely (*Figure 7G*). Furthermore, once silenced, pushing the network back to its active state is also dependent on input ratio (*Figure 7H–J*). However, to make such a transition, the network becomes noticeably more selective about the input ratio (compare *Figure 7G and J*). For both transitions, the proper ratios of the inputs are effectively determined by the approximated initial phase of the network response (*Figure 7D*), and thus mainly dependent on the synaptic weights right before the input arrival. In sum, these results suggest that proper input strengths onto the PC and IN populations, along with the inhibitory action of GABA, play key roles in the dynamic state transitioning of the network, thereby allowing for its burstiness.

Considering the dynamics of synaptic weights in our model along with their significance for state transitions, we next investigated the impact of input timing (*Figure 8*). We found that, once silenced by the first input, a deadline is formed for the network's transitioning back to the active state (dotted line in *Figure 8A, D, G and H*). If the second input misses the deadline, the network will elicit a large-amplitude simNB but is not able to converge to the active state any longer (*Figure 8D*). Prior to this deadline and depending on the input ratio (*Figure 7J*), the network will either transition to the active state (*Figures 8A and 7H*) or return to the silent state (*Figure 7I*). Importantly, our analysis showed that this deadline is an internal property of the network and cannot be overruled by any input level

(see below). Therefore, specific combinations of input ratio (*Figure 7J*) and input timing (*Figure 8G*) are required for transitioning to the active state. In addition, once the simNB failed to converge to the active state, the network will encounter a new deadline (Appendix 1 and *Figure 8—figure supplement 1*). In sum, these results imply that the silent state of the network can have per se different hidden sub-states, each with a specific input-encoding operating scheme.

Having found the intrinsic deadline as a main determinant for the type of NB, we next investigated the origin of these different activity patterns. How does the network decide between transitioning to the active state and returning to the silent state? Remarkably, we found that the deadline for network transitioning to the active state is mechanistically dependent on the presence of a transient stable FP in its fast dynamics around the peak of the simNB. This can be seen in the two examples where the network receives the same input but at different inter-pulse intervals (IPIs), one preceding (*Figure 8A–C*) and the other exceeding the deadline (*Figure 8D–F*). For both IPIs, at the time right before the second input (*Figure 8B and E*), the frozen RNNs only provide evidence for the emergence of simNB, but not for the state transition (note the presence of an amplification domain; pink area). Importantly, we found, however, that in the case of the shorter IPI, the network is able to form a transient, stable non-zero FP in its frozen RNN, at the peak of the simNB (compare *Figure 8C and F*). This FP can transiently attract the network's activity toward itself, and as the activity evolves accordingly, it also changes its position in the corresponding updated frozen STP-RNN, until eventually converging to its counterpart in the full system. Intuitively, this transient, stable FP can guide the network's activity toward that of the full system (see the non-origin green dot in *Figure 7B*). The temporal repositioning of this stable FP is due to the activity-dependency of the synaptic weights in our model. Besides, our findings show that the existence of this FP around the simNB peak is effectively determined by simNB size (*Figure 8G*). If simNB size exceeds an internally determined threshold, the network cannot build such a transient stable FP due to a reduction of synaptic weights (*Rahmati et al., 2017*); consequently, the simNB will be attracted toward the silent state. In this line, *Figure 8G* shows that simNB size is effectively determined by the IPI. The longer the IPI (thus, the silent period) is, the larger the simNB will be. Here, the IPI-dependency of the simNB size mainly reflects the slow recovery from short-term depression of excitatory synapses at the silent state (*Figure 8H*).

In conclusion, our modeling results indicate that developing CA1 possesses multiple input-encoding schemes, which are effectively determined by three factors: (1) the input ratio, (2) the input timing, and (3) the non-linearity and dynamics of synaptic weights.

## A bi-stable STP-RNN model with inhibitory GABA robustly explains our experimental observations

We next investigated whether alternative network models (or mechanisms) are better suited to explain the observed CA1 dynamics in the second postnatal week. By decreasing the population-activity thresholds (θ) and/or changing the polarity of GABAergic synapses from inhibitory to excitatory, we created two operationally distinct models called Mono-RNNi ($\theta_{PC}\downarrow$, 'i' for inhibitory GABA) and Mono-RNNe ($J_{IN} \rightarrow -0.5 \times J_{IN}$, $\theta_{PC}\downarrow$, $\theta_{IN}\downarrow$; 'e' for excitatory GABA; *Figure 9—figure supplement 1A*). Of note, lower θ reflects a higher background input to, or a lower spike threshold of, the neuronal population (see Methods). Each of the models is mono-stable with one spontaneous FP in an active state (non-origin green dots in *Figure 9—figure supplement 1A*).

As a prerequisite for comparison, both Mono-RNNi and Mono-RNNe can generate silent periods, simNBs, and an active state with $A_{PC} > 0$ (*Figure 9—figure supplement 1B–F*), as observed in our data. As compared to the bi-stable STP-RNN model (*Figure 7* and *Figure 9—figure supplement 1A*), however, both mono-stable networks are less plausible in explaining our experimental observations for several reasons. First, due to the lack of a silent FP, Mono-RNNi and Mono-RNNe can transition to and remain in a silent state only in the (continuous) presence of synaptic input (*Figure 9—figure supplement 1C and D*). Second, for silencing ($A_{PC}=A_{IN}=0$), the external input to at least one of the populations needs to be inhibitory (*Figure 9—figure supplement 1E*, green areas). However, input to CA1 at this age is mainly mediated by glutamatergic projections from entorhinal cortex and CA3. In addition, considering inhibitory input violates the assumption of excitatory GABAergic synapses in the Mono-RNNe. Third, both Mono-RNNi and Mono-RNNe require a relatively long silencing (of at least the PC population) to effectively generate simNBs (*Figure 9—figure supplement 1F*), which is difficult to reconcile with the time-course of SPWs (tens of milliseconds).

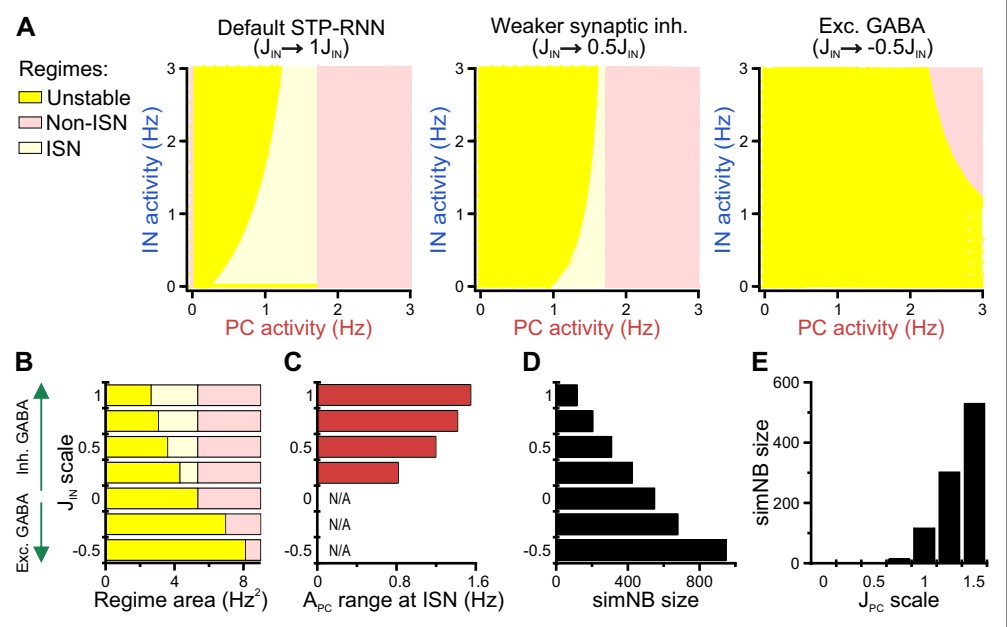

**Figure 9.** Inhibitory stabilization of a persistent active state in the bi-stable short-term synaptic plasticity (STP)-recurrent neural network (RNN) model. (**A**) The inhibition-stabilized network (ISN) regime becomes accessible to the network upon the developmental emergence of synaptic inhibition. The colored regions in each $A_{IN}$-$A_{PC}$-plane of the network model depict the fixed point (FP)-domains of three possible operating regimes: unstable dynamics, ISN, and Non-ISN. Synaptic inhibition strength $J_{IN}$ was set to 3 (inhibitory; left), 1.5 (inhibitory, middle), and –1.5 (excitatory, right). (**B–D**) Effect of strength and polarity of GABAergic synapses ($J_{IN}$) on the availability of the ISN regime (**B**), the maximum range of pyramidal cell (PC) activity in the ISN regime (**C**), and size of simulated network bursts (simNBs) when triggered at the rest state (**D**). Results were obtained by scaling $J_{IN}$ in the STP-RNN model with values indicated on the y-axis. The area of each operating regime was computed based on the area of its FP-domain in the $A_{IN}$-$A_{PC}$-plane (with limits as in A). (**E**) Dependency of simNB size on the strength of glutamatergic synapses ($J_{PC}$). Default value of $J_{PC}$ in STP-RNN was 6.5 (corresponding to a scale of 1).

The online version of this article includes the following figure supplement(s) for figure 9:

**Figure supplement 1.** A bi-stable short-term synaptic plasticity (STP)-recurrent neural network (RNN) model with inhibitory GABA robustly explains the experimental observations.

The intrinsic bi-stability of the STP-RNN renders it computationally different from the mono-stable alternatives: (1) In the STP-RNN, simNBs are triggered by suitable inputs (*Figures 7 and 8*), whereas in Mono-RNNi and Mono-RNNe, simNBs are elicited in the form of rebound bursts only after the cessation of the non-specific silencing inputs (*Figure 9—figure supplement 1C, D and F*). This feature renders the STP-RNN potentially more suitable for input discrimination prior to the onset of environmental exploration. (2) Regardless of their size, rebound simNBs will always return to the active state in Mono-RNNi and Mono-RNNe (*Figure 9—figure supplement 1C and D*), in contrast to the STP-RNN, which may also return to its rest state (*Figures 7 and 8*). (3) Mono-RNNe lacks an inhibition-stabilized network (ISN) regime (*Figure 9—figure supplement 1A*), which is thought to provide a general strategy for supporting more complex computations (*Latham and Nirenberg, 2004*; *Tsodyks et al., 1997*).

In sum, these results indicate that the proposed STP-RNN, operating under bi-stability, is not only computationally more flexible but also more plausible in explaining our experimental observations.

## Inhibitory stabilization of a persistent active state in the bi-stable STP-RNN model

Using our STP-RNN model, we further investigated the functional significance of the network behavior during the second postnatal week. Our analyses show that an ISN regime is effectively accessible at this age due to the developmental increase of synaptic inhibition (*Figure 9A and B*). This presumably enables CA1 to process more complex computations (while avoiding instabilities) in parallel to the

developmental strengthening of sensory inputs. Importantly, the active state in our model is also located in the ISN FP-domain (*Figure 9—figure supplement 1A*, left). Such an ISN regime is absent if GABAergic transmission is excitatory (*Figure 9A* [right] and *Figure 9B*). Moreover, the developmental increase in the strength of synaptic inhibition ($J_{IN}$) enables the network to operate as an ISN at a wider range of PC activity levels (*Figure 9C*). This is because the unstable FP-domain, confined to low levels of PC activity, is progressively replaced by the ISN FP-domain as inhibitory synapses become stronger (*Figure 9A and B*). In the model, simNB size is reduced by a strengthening of inhibition (akin to our experimental observations at P11 vs. P18), but increased if GABA is considered excitatory (unlike our data at P11 vs. P4; *Figure 9D*). This implies that the changes in inhibitory strength alone are unsuited to explain the enhanced burstiness in the second postnatal week. We found, however, that the experimentally reported, concurrent developmental strengthening of glutamatergic synapses ($J_{PC}$; for review see *Kirmse and Zhang, 2022*) exerts an opposite effect by profoundly amplifying simNBs (*Figure 9E*). Collectively, our analyses portend that the emergence of a persistent active state in CA1 reflects the developmental strengthening of both GABAergic inhibition ($J_{IN}$) and glutamatergic excitation ($J_{PC}$) as well as changes in background input and/or intrinsic excitability ($\theta$). In line with this, the enhanced burstiness at P11 is an expression of the complex neural dynamics in a bi-stable STP-RNN that identify the second postnatal week as a key transitional period in CA1 network maturation.

## Discussion
### Developmental trajectories of network dynamics in CA1 in vivo

Our data demonstrate that the activity of CA1 PCs in the first postnatal week is generally low and exclusively discontinuous in nature, as they spend most of the time in a silent rest state that is only interrupted by relatively brief NBs (*Figures 2A–C , and 3A–C*). In other words, CA1 PCs are incapable of sustaining persistent activity at this age, similar to findings in visual cortex (*Ackman et al., 2012*; *Hanganu et al., 2006*; *Kirmse et al., 2015*; *Kummer et al., 2016*). Previous data-driven computational modeling further showed that such dynamics can be emulated by mono-stable network models, in which excitatory GABA effectively promotes network burstiness (*Flossmann et al., 2019*). CA1 NBs differ from their neocortical counterparts in that the latter exhibit a distinct horizontal confinement that appears to be largely absent in CA1. This is evident in the weak distance-dependency of pairwise correlations of CA1 PCs (*Figure 4H*), extending previous results from large-scale imaging (*Graf et al., 2021*) and electrophysiological (*Valeeva et al., 2019*; *Valeeva et al., 2020*) studies. Whether differences in spatial properties of NBs in neocortex vs. hippocampus reflect their specific input characteristics and/or connectivity patterns is an open question. Likewise, the developmental relevance of such differences is unknown. However, wavefront-containing activity patterns have been causally linked to the developmental refinement of topographic maps (*Cang et al., 2005*; *Li et al., 2013*) and receptive field characteristics (*Albert et al., 2008*; *Wosniack et al., 2021*) in the visual cortex, suggesting that their absence in CA1 might reflect the lack of a clear topical macro-organization of the hippocampus (*Bellistri et al., 2013*).

We here identify the second postnatal week as a key transitional period in CA1 network maturation. For the first time in development, CA1 PCs are able to maintain spontaneous persistent activity, while this transition toward embedding continuous activity is largely completed by P18 (*Figure 3A–C*). In the second postnatal week, when discontinuous activity still dominates (*Figure 3C*), CA1 PCs undergo a transient period of enhanced network burstiness (*Figure 3A*). This trajectory markedly differs from what has been previously reported for the hippocampus in vitro, where GDPs disappear shortly after the first postnatal week. At this time, GABA-releasing INs already inhibit CA1 PCs (*Murata and Colonnese, 2020*; *Spoljaric et al., 2017*; *Tyzio et al., 2008*), implying that NBs in vivo do not depend on GABAergic excitation (in contrast to GDPs). NBs are also observed at P18, but these are short in duration (*Figure 3G*) and recruit fewer neurons (*Figure 3H*) than at earlier stages. Strikingly, whereas NBs at P11 are large and frequent, repetition rates of recurring cellular activation patterns (a prime characteristic of adult hippocampal activity) significantly increased only after the onset of environmental exploration (*Figure 5D*). This was accompanied by highly skewed firing rate distributions at P18 (quantified as CaT frequencies; *Figure 2B–E*). The latter are thought to underlie sparse coding (*Ikegaya et al., 2013*; *Narayanan and Johnston, 2012*; *Roxin et al., 2011*; *Trojanowski et al., 2020*; *Yassin et al., 2010*), an energy-efficient regime of input processing and information storage (*Mizuseki*

*and Buzsáki, 2013*). Collectively, our data indicate that CA1 network activity acquires a number of 'adult-like' characteristics by P18, i.e., shortly after the onsets of pattern vision, active whisking, and environmental exploration. At P11, prominent network burstiness and emergent continuity were also observed in unanesthetized mice (*Figure 6*), confirming that they did not result from the use of nitrous oxide. However, further investigations are warranted, as potential age- and state-dependent effects of nitrous oxide on neuronal dynamics are currently unknown. This might be particularly relevant in the context of the emergence of an active sleep-wake cycle around eye opening (see also *Chini et al., 2019*; *Shen and Colonnese, 2016*).

## A role for intrinsic network instability and synaptic inhibition in NB generation in CA1

During the transition period (i.e. P11), bursting activity exhibited a preferred frequency of ~0.1–0.5 Hz, indicating that NBs occur in a temporally non-random manner (*Figure 3*). However, individual neurons were recruited more randomly at this age, as the number of significant motifs of network activity as well as their average repetition probability were lowest (*Figure 5*). At P11, the occurrence of CaTs in individual cells resembled a Poissonian process (*Figure 2*), and pairwise neuronal correlations were lower than at P4 (*Figure 4G*). We here set out to explain these seemingly discordant experimental findings using data-informed computational modeling.

Capitalizing on a dynamic systems modeling approach, we show that a potential dynamical regime of the network that allows for the generation of NBs in the presence of effective synaptic inhibition is bi-stability. We found that our network model is prone to an intrinsic instability, governed by a nonlinear interaction between its fast (firing) and slow (synaptic) dynamics. Such instability enables the model to over-amplify the input, even after its removal, and thus elicit NBs (simNBs; *Figure 7*). This indicates that a (sim)NB reflects a spatiotemporal trajectory of the network's intrinsic instability dynamics, which, due to its nature, can recruit a random set of cells at random order within a specific time-window (*Rahmati et al., 2017*). Thus, the data-informed model mechanistically links strong PopC to weak pairwise neuronal correlations, the close-to-random firing of individual PCs, and the low number of network motifs – as we found experimentally for the second postnatal week.

What are the functional roles of burstiness and synaptic inhibition at this stage? Our model, in addition to its silent state, embeds a stable FP (or steady state) at non-zero low activity rates (*Figure 7B*), in accordance with our recorded data. Theoretical studies indicate that the presence of such an FP requires the stabilization function of inhibitory GABA (*Latham and Nirenberg, 2004*; *Ozeki et al., 2009*; *Rahmati et al., 2017*; *Tsodyks et al., 1997*). We here show that, at such a FP, the analyzed network model operates under an ISN regime, which may enable CA1 networks to begin performing complex computations (*Latham and Nirenberg, 2004*; *Tsodyks et al., 1997*). We also show that the ISN FP-domain becomes effectively expanded by the strengthening of inhibition, i.e., a larger set of stimulus-evoked ISN attractors (or FPs) are accessible for network computations, in parallel to the developmental strengthening of sensory inputs. Therefore, elucidating how GABAergic INs contribute to NBs and emergent continuity is a promising objective for future experimental studies, which could also constrain computational models of developing CA1. The ability of the network to dynamically transition between its silent and active states in an input-dependent fashion (*Figure 7*) renders the second postnatal week an early developmental stage toward forming hippocampal memory and cognition mechanisms, as found in adult hippocampal attractor networks (*Hartley et al., 2014*; *Knierim and Neunuebel, 2016*; see also *Rahmati et al., 2017*; *Rolls, 2007*). This view is supported by (1) the existence of the internal deadlines as well as a delicate input-ratio and -timing dependency of successful state transitions and simNB generation and (2) the network's ability to store information in both the silent and active state through transient synaptic weights (*Barak and Tsodyks, 2014*; *Mongillo et al., 2008*; *Stokes, 2015*) and persistent activity (*Boran et al., 2019*; *Zylberberg and Strowbridge, 2017*). Our modeling results further imply that the network's silent state has per se several dynamic operational sub-states, which keep track of input timing and strength (*Figure 8* and *Figure 8—figure supplement 1*) to produce proper network read-outs. Collectively, we postulate that the basis of CA1 encoding schemes is set in shortly before eye opening. Moreover, our data suggest that GDPs disappear because synaptic input ratios required for NB generation (*Figure 7J*) are not preserved in in vitro preparations.

## Potential developmental functions of NBs in the neonatal CA1

Computational modeling suggests a mechanism, whereby CA1 undergoes extensive input-discrimination learning before eye opening. In this scenario, NBs serve as a feedback that informs individual CA1 PCs about functionally important characteristics of the synaptic input to the local network, including (1) the proper targeting ratio of excitatory PCs vs. inhibitory GABAergic INs (*Murata and Colonnese, 2020*; *Valeeva et al., 2016*) and (2) the timing of inputs relative to the network's operational state. Interestingly, the developmental period of enhanced network burstiness coincides with a major surge of synaptogenesis in CA1 PCs (*Kirov et al., 2004*). The latter involves a net addition of synapses, but also functionally important anatomical rearrangements. Specifically, the formation of mature dendritic spines, which allow for electrical and metabolic compartmentalization of postsynaptic responses, commences only at around P10, by which time most glutamatergic synapses are rather localized to dendritic shafts (*Fiala et al., 1998*; *Kirov et al., 2004*). In addition to acting as potential synaptogenic stimuli (*Kirov et al., 2004*), NBs could thus be an important element underlying synaptic competition and pruning, i.e., based on synchronization-dependent plasticity rules in nascent dendrites (*Winnubst et al., 2015*). Network burstiness might therefore be causally related to the delayed development of skewed (approximately log-normal) firing rate distributions (*Figure 2*) underlying sparse coding (*Ikegaya et al., 2013*; *Narayanan and Johnston, 2012*; *Roxin et al., 2011*; *Trojanowski et al., 2020*; *Yassin et al., 2010*). In accordance with the efficient coding hypothesis and seminal work in the visual system (*Albert et al., 2008*), we argue that one function of developing CA1 and, thus, NBs is to remove statistical redundancy in the multi-sensory place-field code, by making use of a learning scheme that uses both intrinsically and sensory-evoked activity already before environmental exploration.

## Methods

**Key resources table**

| Reagent type (species) or resource | Designation | Source or reference | Identifiers | Additional information |
|---|---|---|---|---|
| Strain and strain background (*Mus musculus*) | B6.129S2-*Emx1*$^{tm1(cre)Krj}$/J (*Emx1*$^{IREScre}$) | The Jackson Laboratory | RRID: IMSR_JAX:005628 | |
| Strain and strain background (*Mus musculus*) | B6;129S6-*Gt(ROSA)26Sor*$^{tm96(CAG-GCaMP6s)Hze}$/J (*Rosa26*$^{LSL-GCaMP6s}$) | The Jackson Laboratory | RRID: IMSR_JAX:024106 | |
| Software and algorithm | Wolfram Mathematica 13 | Wolfram | RRID:SCR_014448 | |
| Software and algorithm | Matlab 2021b | Mathworks | RRID:SCR_001622 | |
| Software and algorithm | Fiji | PMID:22743772 | RRID:SCR_002285 | |
| Software and algorithm | *Calcium transient detection harnessing spatial similarity* (CATHARSiS) | This paper | N/A | https://github.com/kirmselab/CATHARSiS |

### Animals

All animal procedures were performed with approval of the local government (Thüringer Landesamt für Verbraucherschutz, Bad Langensalza, Germany; reference no.: 02-012/16) and complied with European Union norms (Directive 2010/63/EU). Animals were housed in standard cages with 14 hr/10 hr light/dark cycles. *Emx1*$^{IREScre}$ (strain #: 005628) and *Rosa26*$^{LSL-GCaMP6s}$ (strain #: 024106) mice were originally obtained from the Jackson Laboratory. Double heterozygous offspring (*Emx1*$^{IREScre}$:*Rosa26*$^{LSL-GCaMP6s}$ mice) was used for experiments at P3–4 ('P4'), P10–12 ('P11'), and P17–19 ('P18'). Mice of either sex were used.

### Surgical preparation, anesthesia, and animal monitoring for in vivo imaging

30 min before starting the preparation, 200 mg/kg metamizol (Novacen) was subcutaneously injected for analgesia. Animals were then placed onto a warm platform and anesthetized with isoflurane (3.5%

for induction and 1–2% for maintenance) in pure oxygen (flow rate: 1 l/min). The skin overlying the skull was disinfected and locally infiltrated with 2% lidocaine (s.c.) for local analgesia. Eyes of P17–19 mice were lubricated with a drop of eye ointment (Vitamycin). Scalp and periosteum were removed, and a custom-made plastic chamber with a central borehole (Ø 2.5–4 mm) was fixed on the skull using cyanoacrylate glue (Uhu; P4: 3.5 mm rostral from lambda and 1.5 mm lateral from midline; P11: 3.5 mm rostral from lambda and 2 mm lateral from midline; P18: 3.5 mm rostral from lambda and 2.5 mm lateral from midline).

For the hippocampal window preparation (*Mizrahi et al., 2004*), the plastic chamber was tightly connected to a preparation stage and subsequently perfused with warm artificial CSF (ACSF) containing (in mM): 125 NaCl, 4 KCl, 25 NaHCO$_3$, 1.25 NaH$_2$PO$_4$, 2 CaCl$_2$, 1 MgCl$_2$, and 10 glucose (pH 7.4, 35–36°C). A circular hole was drilled into the skull using a tissue punch (outer diameter 1.8 mm for P4 and 2.7 mm for P11 and P18 mice). The underlying cortical tissue and parts of corpus callosum were carefully removed by aspiration using a vacuum supply and a blunt 27 G or 30 G needle. Care was taken not to damage alveus fibers. As soon as bleeding stopped, the animal was transferred to the microscope stage.

During in vivo recordings, body temperature was continuously monitored and maintained at close to physiological values (36–37°C) by means of a heating pad and a temperature sensor placed below the animal. Spontaneous respiration was monitored using a differential pressure amplifier (Spirometer Pod and PowerLab 4/35, ADInstruments). Isoflurane was discontinued after completion of the surgical preparation and gradually substituted with the analgesic-sedative nitrous oxide (up to the fixed final N$_2$O/O$_2$ ratio of 3:1, flow rate: 1 l/min). Experiments started 60 min after withdrawal of isoflurane. At the end of each experiment, the animal was decapitated under deep isoflurane anesthesia.

In a separate set of experiments (*Figure 6*), we analyzed the effects of N$_2$O on animal state and CA1 network dynamics at P10–12. To this end, the following experimental timeline was applied in an additional cohort of six mice (*Figure 6—figure supplement 2A*). In the first FOV per mouse, Ca$^{2+}$ imaging started under N$_2$O (N$_2$O/O$_2$ ratio of 3:1 as above). About 10 min after replacing N$_2$O by pure O$_2$ (unanesthetized), Ca$^{2+}$ imaging was continued in the same FOV (i.e. from the same cells). We then moved to a second FOV (with another set of cells) and performed recordings in a reversed order, i.e., Ca$^{2+}$ imaging started under unanesthetized conditions before switching to N$_2$O/O$_2$ (recordings started 10 min after the onset of N$_2$O administration). We reduced laser power and increased detector gain (as compared to recordings presented in *Figures 2–5*) to prevent photo-bleaching and -toxicity in these longer-lasting experiments.

## Two photon Ca$^{2+}$ imaging in vivo

After transferring the animal to the microscope stage, ACSF was removed, and the hippocampal window was filled up with a droplet of agar (1%, in 0.9% NaCl) and covered with a cover glass. As soon as the agar solidified, the chamber was again perfused with ACSF.

Imaging was performed using a Movable Objective Microscope (Sutter Instrument) equipped with two galvanometric scan mirrors (6210 H, MicroMax 673 XX Dual Axis Servo Driver, Cambridge Technology) and a piezo focusing unit (P-725.4CD PIFOC, E-665.CR amplifier, Physik Instrumente) controlled by a custom-made software written in LabVIEW 2010 (National Instruments; *Kummer et al., 2015*) and MPScope (*Nguyen et al., 2006*). Fluorescence excitation at 920 nm was provided by a tunable Ti:Sapphire laser (Chameleon Ultra II, Coherent) using a 20×/1.0 NA water immersion objective (XLUMPLFLN 20XW, Olympus). Emission light was separated from excitation light using a 670 nm dichroic mirror (670 DCXXR, Chroma Technology), short-pass filtered at 680 nm, and detected by a photomultiplier tube (12 bit, H10770PA-40, Hamamatsu). Data were acquired using two synchronized data acquisition devices (NI 6110, NI 6711, National Instruments). Sampling rate was set to 11.63 Hz (256×256 pixels, 248×248 µm). For each animal, spontaneous activity was recorded from 3 to 5 FOVs, each one usually for ~20 min. Some FOVs were excluded from further analysis due to excessive z-drifts. Finally, 1–4 FOVs were analyzed per animal and used for statistics. Any spatial overlap between sequentially recorded FOVs was strictly avoided based on xyz-coordinates of the objective and visual control.

## Quantification and statistical analysis

### Preprocessing

Image stacks were registered using NoRMCorre (*Pnevmatikakis and Giovannucci, 2017*). For residual drift detection, a supporting metric was calculated as the Pearson correlation coefficient of the binarized template image used for stack registration and the binarized images of the registered image stack. Time periods with residual drift were then visually identified (by inspecting the supporting metric and the aligned image stack) and considered as missing values in subsequent analyses. Raw ROIs were manually drawn around the somata of individual CA1 PCs using Fiji.

We quantified $\Delta F/F_0$ noise levels as the mean (per cell) difference between the 50th and the 10th percentile of the $\Delta F/F_0$ distribution. Noise levels were similar across the age groups (#5 in *Supplementary file 1b*).

### Calcium transient detection harnessing spatial similarity

For the detection of CaTs in densely labeled tissue, we devised CATHARSiS. CATHARSiS makes use of the fact that spike-induced somatic GCaMP signals ($\Delta F$) are spatially non-uniform and characteristic of a given cell. CATHARSiS comprises three major steps: (1) the generation of a spatial $\Delta F$ template representing the active cell, (2) the computation of a detection criterion $D(t)$ for each time point (frame), and (3) the extraction of CaT onsets. All analyses were performed using custom scripts in Matlab and Fiji. CATHARSiS is available via GitHub (https://github.com/kirmselab/CATHARSiS).

Ad (1): for each ROI, we first obtained the mean F(t) by frame-wise averaging across all pixels of that ROI. We then computed the first derivative of F(t) and smoothed it using a second order Savitzky-Golay algorithm (window length, six frames), thus yielding Ḟ(t). We then determined eight candidate CaT onsets by extracting the frame numbers corresponding to the eight Ḟ(t) peaks having the largest amplitude. This step was performed in an iterative-descending manner by starting with the largest F(t) peak. For each peak, we defined a minimum time difference (five frames) to all subsequently extracted peaks, so as to avoid extracting nearby frames belonging to the same CaT. For each candidate CaT onset, we then computed the corresponding spatial $\Delta F$ (average of five successive frames). To this end, we first radially expanded the raw ROI by two pixels using the Euclidian distance transform (we found that this increased detection reliability due to enhanced spatial contrast). Resting fluorescence $F_0(t)$ was defined as the moving median over 500 frames. Eight candidate $\Delta F$ templates were obtained by converting raw $\Delta F$ values into z-scores. Based on visual inspection, we next rejected those candidate $\Delta F$ templates that putatively reflected activation of optically overlapping somata and/or neurites. If all candidate $\Delta F$ templates had been rejected, the cell was excluded from further analysis; otherwise, the remaining candidate $\Delta F$ templates were averaged to obtain the final $\Delta F$ template representing the active cell.

Ad (2): for each ROI (spatially expanded as above), we extracted its spatial $\Delta F$ for all frames in the image stack. Next, the spatial $\Delta F$ template representing the active cell was optimally scaled to fit its $\Delta F$ in each recorded frame. Based on the optimum scaling factor and the goodness of the fit, a detection criterion $D(t)$ was computed for each time point. Here, $D(t)$ was defined without modification as previously described for the temporal domain (*Clements and Bekkers, 1997*).

Ad (3): for each ROI, CaT onsets were extracted from $D(t)$ using UFARSA (*U*ltra-*f*ast *a*ccurate *r*econstruction of *s*piking *a*ctivity), a general-purpose event detection routine (*Rahmati et al., 2018*). To this end, we slightly modified the original UFARSA approach in two ways. (1) Following the smoothing step implemented in UFARSA, all negative values were set to zero, as we found in our preliminary analysis that negative-to-positive transitions occasionally resulted in false positive events. (2) We introduced a lower bound for the leading threshold, so as to minimize potential false positive events. Reconstructed CaT onsets were translated into a binary activity vector (1 – event, 0 – no event) and used for the following analyses.

For the analyses shown in *Figure 1E–I*, we compared CATHARSiS to an algorithm based on mean $\Delta F(t)$. To this end, for a given ROI, we first computed $\Delta F(t)$ by frame-wise averaging over all pixels belonging to that ROI. We then extracted CaT onsets from $\Delta F(t)$ using UFARSA, a general-purpose event detection routine (*Rahmati et al., 2018*).

## Firing irregularity

For each cell, we quantified the irregularity of its CaT onsets (i.e. firing times) using CV2, as a local and relatively rate-independent measure of spike time irregularity (*Holt et al., 1996*; *Ponce-Alvarez et al., 2010*): $\mathrm{CV2} = \frac{1}{K-1} \sum_{k=1}^{K-1} \frac{2|ICI_{k+1} - ICI_k|}{ICI_{k+1} + ICI_k}$, where $ICI_k$ and $ICI_{k+1}$ are the $k$th and $(k{+}1)$th inter-CaT intervals of the cell, and $K$ is the total number of its $ICI$s. To achieve more robust results, cells with less than 10 $ICI$s were excluded from this analysis.

## Network bursts

NBs were defined as a significant co-activation of cells as follows: (1) To account for some temporal jitter in the detection of CaT onsets, all values in the binary activity vectors that fell within $\pm\Delta t$ frames of any detected CaT were set to 1. Unless otherwise stated, $\Delta t$ was set to 3. We then computed the mean across the resulting activity vectors of all individual cells to obtain the empirical fraction of active cells per frame $\Phi(t)$. (2) We randomly shuffled CaT onsets of all cells (uniform distribution; 1000 times), computed the surrogate $\Phi(t)$ (as above), and defined the 99.99th percentile of all surrogate $\Phi(t)$ as the threshold for NB detection. The NB threshold was determined separately for each FOV, so as to account for different mean CaT frequencies. (3) Any frame with an empirical $\Phi(t)$ exceeding the threshold was considered as belonging to an NB.

To examine the robustness of our findings, we systematically varied the operational definition of the threshold used for NB detection in two ways. (1) In the first approach, $\Delta t$ was set to values ranging from 1 to 11 frames. A frequency-dependent threshold was then computed for each FOV as detailed above (*Figure 3—figure supplement 1A*). (2) In the second approach, a constant (frequency-independent) threshold was applied to all FOVs (ranging from 7 to 17% active cells per frame) (*Figure 3—figure supplement 1B*). Here, $\Delta t$ was set to three frames. Note that the threshold values below ~10% are less meaningful, as the average fraction of active cells in some FOVs at P18 is ~9%.

In the resulting binary NB vectors, 0–1 transitions were defined as NB onsets and 1–0 transitions as NB offsets. Using the binary NB vectors, we extracted (1) the relative time the network spent in NBs and (2) the average NB duration. NB size was defined as the fraction of cells which were active in at least one frame of a given NB, corrected for the chance level of co-activation by subtracting the NB threshold.

## Discontinuous and continuous network activity

We operationally defined periods of discontinuous or continuous network activity as follows: we partitioned recordings into non-overlapping time bins of 116 frames (~10 s) duration. Network activity during a given time bin was classified as continuous if the fraction of active cells per frame $\Phi(t)$ exceeded 3% in >70% of all frames belonging to that bin; if $\Phi(t)$ exceeded 3% in ≤70% of all frames belonging to that bin, it was considered discontinuous. To compute $\Phi(t)$, $\Delta t$ was set to three frames. Note that it is currently unknown how $\Phi(t)$-based (dis-)continuity correlates with (dis-)continuity observed in local-field potential data.

## Power analysis

To account for missing values representing the residual drift periods (see above), spectral power of the fraction of active cells $\Phi(t)$ was estimated by computing the Lomb-Scargle periodogram (Matlab, MathWorks). To compute $\Phi(t)$, $\Delta t$ was set to three frames.

## Pairwise correlations

STTCs were computed for all possible cell pairs with a synchronicity window $\Delta t$ of three frames (~258 ms) using custom written code (Matlab, MathWorks; *Cutts and Eglen, 2014*). STTCs derived from measured data were compared to those from surrogate data obtained by randomly shuffling (uniform distribution; 1000 times) CaT onsets of all cells, separately. This randomization kept the mean CaT frequency of each cell unchanged.

## Population coupling

To quantify the degree of coupling of each cell to the overall population activity, we computed its PopC (*Okun et al., 2015*; *Sweeney and Clopath, 2020*). To this end, for each cell, we first smoothed

its binary vector (see above) and the summed vector of the rest of the population, followed by computing PopC as the Pearson correlation coefficient between these two vectors. For smoothing, we used a Gaussian kernel with SD = 3 frames. To assess the significance of the PopCs (i.e. being beyond chance), we generated surrogate data by binning the raster matrix along time-axis; non-overlapping bins with a size of 10 frames (ca. $\sqrt{12}SD$, according to *Kruskal et al., 2007*). We randomly exchanged CaT onsets across active cells within each bin (500 times), thereby effectively preserving the CaT frequency of each cell as well as the local summed activity of the population. For each cell, using its surrogates, we determined the significance of its empirical PopC (95th percentile). Moreover, when reporting the PopC of each cell, we subtracted the mean of its surrogate PopCs in order to account for the potential differences in population activity levels of different FOVs (for a similar approach see *Okun et al., 2015*; *Sweeney and Clopath, 2020*). Cells with less than five CaTs were excluded from this analysis to increase robustness of our results.

## Motifs of population activity

To identify the specific cellular activation patterns recurring over time (i.e. motifs of population activity), we used an eigendecomposition-based clustering method (*Li et al., 2010*; *Patel et al., 2015*). To this end, we first divided the recording time into non-overlapping windows with a size of 10 frames and assigned 1 and 0 to cells which were active or silent during each bin. This converts the raster matrix to a sequence of binary vectors (i.e. spatial patterns), where each pattern has a size of Nx1 (N is the number of analyzed cells in the FOV). We then computed the degree of similarity between all possible pairs of these patterns using matching index (MI, *Romano et al., 2015*): $\mathrm{MI}_{ij} = 2\frac{|Pat_i \cap Pat_j|}{|Pat_i| + |Pat_j|}$, where $Pat_i$ and $Pat_j$ are the $i$th and $j$th binary cellular activation patterns (vectors), and the norms are equal to the number of ones (i.e. active cells) in each vector. MI ranges from 0 (no similarity) to 1 (perfect similarity), and in particular approximates the number of common neuronal activations (i.e. common ones) between pattern pairs; for more details see *Romano et al., 2015*; *Sporns et al., 2007*. Accordingly, for each FOV, we obtained a similarity matrix of size P × P, where P indicates the number of patterns. The rows and columns relating to the silent-pattern pairs were excluded, as they were giving rise to an undefined value (i.e. 0 divided by 0). We used the MI matrix as the input to the eigendecomposition clustering method. Briefly, this method decomposes a given similarity matrix (here, MI matrix) into a set of eigenvalues and eigenvectors. The number of significantly large eigenvalues determines the number of motifs, and their corresponding eigenvectors contain the information about motif structure (i.e. the set of patterns belonging to each motif). The largest eigenvalue is proportional to the global similarity among all patterns. As the surrogate data for testing the statistical significance of the eigenvalues and also computing a normalized unbiased value of global similarity index, we used the randomly shuffled CaT onsets (see above), based on which we repeated the binning and computation of MI matrices (500 times). This procedure enabled us to identify the motifs of cellular activation patterns, which occurred beyond chance level. For more details about the clustering method and its mathematical description see *Li et al., 2010*.

## Analysis of cardiovascular parameters and movement periods

Respiration and movement were recorded by means of an air pillow positioned below the chest of the animal and a differential pressure amplifier (Spirometer Pod and PowerLab 4/35, ADInstruments). We first computed the short-time Fourier transform using a time window of 1 s with an overlap of 50% (*Figure 6—figure supplement 1*). To extract respiration and heart rates, the median power spectral density was calculated across time, and the first two peaks were detected. In one FOV, this was not possible, as the two peaks could not be reliably separated. For the detection of movement periods, bandpower (0.1–8 Hz) was first calculated across time. A time bin was classified as movement, if the bandpower exceeded a threshold, defined as the moving median (over 60 s) plus three times the moving absolute deviation (over 300 s). In the resulting binary movement vectors, 0–1 transitions were defined as movement onsets and 1–0 transitions as movement offsets.

# Computational modeling of a developing neural network with inhibitory GABA

## Overview

To gain insights into the mechanisms and functional role of the observed network burstiness during the emergence of synaptic inhibition in CA1, we used computational modeling and stability analysis. For this purpose, we employed a recently established model of an RNN for first postnatal month development (*Rahmati et al., 2017*). It is an extended Wilson-Cowan-type model (*Tsodyks et al., 1998*) and benefits from being biophysically interpretable and mathematically accessible. Recently, this model was also adapted successfully to explain key dynamics and mechanisms of GDPs in neonatal CA1 with excitatory GABA signaling during the first postnatal week (*Flossmann et al., 2019*). However, in accordance with previous reports and our present experimental data for the second postnatal week, we here use the model with mainly two specific cellular properties: (1) GABAergic synapses are considered inhibitory (*Kirmse et al., 2015*; *Murata and Colonnese, 2020*; *Valeeva et al., 2016*) and (2) the mean spontaneous firing activity of PCs is effectively non-zero (*Figure 2C*). In the following, after providing the mathematical description of the model, we describe the mathematical components used for its stability analysis. For more details about the model and the approach see *Rahmati et al., 2017*.

## Model description

The model is a mean-field network model of mean firing activity rates of two spatially localized, homogeneous glutamatergic and GABAergic cells (here, PC and IN populations) that are recurrently connected (*Figure 7A*). The model incorporates two STP mechanisms, namely short-term synaptic depression (STD) and facilitation (STF), which render the synaptic efficacies dynamic over time. Hence, we call the network hereafter STP-RNN. The equations governing the mean-field dynamics of the STP-RNN (10D) are (dots denote the time derivatives and, hereafter, PC and IN are abbreviated as P and I for readability; *Rahmati et al., 2017*):

$$\tau_P \dot{A}_P(t) = -A_P(t) + f_P(J_{PP}u_{PP}(t)x_{PP}(t)A_P(t) - J_{PI}u_{PI}(t)x_{PI}(t)A_I(t) + e_P(t)) = -A_P(t) + f_P(h_P)$$

$$\tau_I \dot{A}_I(t) = -A_I(t) + f_I(J_{IP}u_{IP}(t)x_{IP}(t)A_P(t) - J_{II}u_{II}(t)x_{II}(t)A_I(t) + e_I(t)) = -A_I(t) + f_I(h_I)$$

$$\dot{x}_{ij} = \tau_{r_{ij}}^{-1}(1 - x_{ij}(t)) - u_{ij}(t)x_{ij}(t)A_j(t)$$

$$\dot{u}_{ij} = \tau_{f_{ij}}^{-1}(U_{ij} - u_{ij}(t)) + U_{ij}(1 - u_{ij}(t))A_j(t)$$

(1)

where i and j $\in \{P,I\}$, and j is the index of the presynaptic population, $A_P$ and $A_I$ are the average activity rates (in Hz) of PC and IN populations which can be properly scaled to represent locally the average recorded activities in these populations, $x_{ij}$ and $u_{ij}$ are the average dynamic variables of STD and STF mechanisms, $\tau_P$ and $\tau_I$ are approximations to the decay time constants of the glutamatergic and GABAergic postsynaptic potentials, $\tau_{r_{ij}}$ is the synaptic recovery time constant of depression, $\tau_{f_{ij}}$ is the synaptic facilitation time constant, $U_{ij}$ is analogous to the synaptic release probability, $J_{ij}$ is the average maximum absolute synaptic efficacy of recurrent (i=j) or feedback (i $\neq$ j) connections, and $e_P$ and $e_I$ are the external inputs received by the PC and IN populations from other brain regions or stimulation. In this work, we set the inputs to zero (for spontaneous baseline activity) or model them as excitatory pulse (with variable positive amplitude) with a duration of 20 ms, thereby emulating, e.g., the SPW-driven inputs to the PC and IN populations (*Karlsson et al., 2006*). The transformation from the summed input to each population, $h_i$, to an activity output (in Hz) is governed by the response function, $f_i$, defined as:

$$f_i(h_i) = \begin{cases} 0 & \text{for } h_i \leq \theta_i \\ G_i(h_i - \theta_i) & \text{for } \theta_i < h_i \end{cases}$$

(2)

where $\theta_i$ is the population activity threshold and $G_i$ is the linear input-output gain above $\theta_i$. In this work, we parameterize the STP-RNN as a network model representing mainly a stage during the second postnatal week. To do this, we mainly followed *Rahmati et al., 2017* by setting $\tau_P = 0.015$ s, $\tau_I = 0.0075$ s, $J_{PP} = J_{IP} = J_P = 6.5$, $J_{II} = J_{PI} = J_I = 3$, $\tau_{r_{PP}} = \tau_{r_{IP}} = \tau_{r_P} = 3$ s, $\tau_{r_{II}} = \tau_{r_{PI}} = \tau_{r_I} = 2.5$ s, $\tau_{f_{PP}} = \tau_{f_{IP}} = \tau_{f_P} = 0.4$ s, $\tau_{f_{II}} = \tau_{f_{PI}} = \tau_{f_I} = 0.4$ s, $U_{PP} = U_{IP} = U_P = 0.8$, $U_{II} = U_{PI} = U_I = 0.8$, $\theta_P = 0.22$,

$\theta_\mathrm{I} = 0.53$, $G_\mathrm{P} = G_\mathrm{I} = 1$, and $e_\mathrm{P} = e_\mathrm{I} = 0$ Hz (for spontaneous baseline activity). According to these parameter values: (1) both glutamatergic and GABAergic connections will act depressing; (2) the network will spontaneously have, in addition to a silent state, an active state where both $A_\mathrm{I}$ and, in particular, $A_\mathrm{P}$ are effectively non-zero; and (3) GABAergic transmission will be inhibitory (note the positive value of $J_\mathrm{I}$). Note that points (2) and (3) render the model inherently different from the neonatal STP-RNN used by *Flossmann et al., 2019*. The chosen synaptic efficacies account for the fact that PCs constitute ~90% of the total neuronal population in CA1, despite the relatively weak neuron-to-neuron anatomical connectivity between CA1 PCs (*Bezaire et al., 2016*).

## Frozen STP-RNN

A frozen STP-RNN is obtained by freezing the synaptic efficacies of a STP-RNN, i.e., by fixing the STP variables $x_\mathrm{ij}$ and $u_\mathrm{ij}$ at the values of interest. This will convert the STP-RNN (10D; see *Equation 1*) effectively to a 2D network with constant synaptic weights. As shown in *Rahmati et al., 2017* and *Flossmann et al., 2019*, the frozen STP-RNN can provide a reliable approximation to the stability behavior of an STP-RNN at the state chosen for freezing (see below). The equations governing the dynamics of a frozen STP-RNN are:

$$\tau_\mathrm{P}\dot{A}_\mathrm{P}(t) = -A_\mathrm{P}(t) + f_\mathrm{P}\left(J_\mathrm{PP}^\mathrm{frz}A_\mathrm{P}(t) - J_\mathrm{PI}^\mathrm{FP}A_\mathrm{I}(t) + e_\mathrm{P}(t)\right)$$
$$\tau_\mathrm{I}\dot{A}_\mathrm{I}(t) = -A_\mathrm{I}(t) + f_\mathrm{I}\left(J_\mathrm{IP}^\mathrm{frz}A_\mathrm{P}(t) - J_\mathrm{II}^\mathrm{FP}A_\mathrm{I}(t) + e_\mathrm{I}(t)\right)$$

$$\tag{3}$$

where $J_{ij}^{frz} = J_{ij}u_{ij}^{frz}x_{ij}^{frz}$, and $u_{ij}^{frz}$ and $x_{ij}^{frz}$ are the values of $u_\mathrm{ij}$ and $x_\mathrm{ij}$ (see *Equation 1*) at the state of interest; here, at a silent state, active state, or the time of NB's peak (see Results).

## Phase plane

To visualize the stability behavior of our network model, we used the phase plane analysis based on the activity rates: $A_\mathrm{I}$-$A_\mathrm{P}$-plane (2D). The $A_\mathrm{I}$-$A_\mathrm{P}$-plane sketch includes the curves of the $A_\mathrm{P}$-nullcline and $A_\mathrm{I}$-nullcline representing sets of points for which $\dot{A}_\mathrm{P}(t) = 0$ and $\dot{A}_\mathrm{I}(t) = 0$. Any intersection of these nullclines is called an FP, with the stability needed to be determined (see below). For the STP-RNN, these FPs represent the steady states of the full network, i.e., the 10D STP-RNN in *Equation 1* (see also *Figure 7B*). For the frozen STP-RNN (thus, 2D; see *Equation 3*) with synaptic efficacies frozen at the state of interest (e.g. silent state), these FPs may include that state and possibly some other FPs which may not exist in the STP-RNN itself (e.g. see *Figures 7D and 8C*). In addition to the visualization of the FPs in the $A_\mathrm{I}$-$A_\mathrm{P}$-plane, we also computed the FPs by numerically solving *Equation 1* and *Equation 3* (separately) after setting the right hand side of the equations to zero. For more details see *Rahmati et al., 2017*.

## Stability of FPs

To determine the stability of any FP in the STP-RNN (resp. in the frozen STP-RNN), we applied the linear stability analysis to its 10D (resp. 2D) system of equations in *Equation 1* (resp. *Equation 3*). We investigated whether all eigenvalues of the corresponding Jacobian matrix have strictly negative real parts (if so, the FP is stable), or whether at least one eigenvalue with a positive real part exists (if so, the FP is unstable).

## Simulations

All simulation results in this paper have been implemented as Mathematica and Matlab (MathWorks) code. For network simulations, we set the integration time-step size to 0.0002 s. In *Figure 7C* and *Figure 9—figure supplement 1B–E*, the initial conditions of the STP-RNN variables were set to those values of the spontaneous stable FP of the network at the active state.

## Operating regimes and FP-domains

The stable operating regimes of an RNN at an FP can be classified as an ISN vs. a Non-ISN (*Latham and Nirenberg, 2004*; *Ozeki et al., 2009*; *Rahmati et al., 2017*; *Tsodyks et al., 1997*). To apply this theoretical classification to the STP-RNN model, we used the previously described analytical findings and numerical techniques (for details see *Rahmati et al., 2017*). In brief, to discriminate between

these two regimes in STP-RNN with inhibitory GABAergic synapses, three criteria were defined: (A) excitatory instability: for the inhibitory activity rate fixed at the FP, the recurrent excitation is strong enough to render the PC-population intrinsically unstable. (B) Excitatory stability: in contrast to (A), the PC-population is stable per se, i.e., even with a feedback inhibition fixed at its level at the FP. (C) Overall stability: the dynamic feedback inhibition to the PC-population is strong enough to stabilize the whole network activity. At an FP, a network operating under the (A) and (C) criteria is an ISN, while a network operating under the (B) and (C) criteria is a Non-ISN. A network, which is neither ISN nor Non-ISN at the FP, operates under an unstable regime. Clearly, for the network with excitatory GABAergic synapses, the ISN regime cannot be defined. Therefore, at an FP, the network is either unstable or Non-ISN. However, as in this case the condition (C) is not applicable, the Non-ISN refers to a non-unstable regime, i.e., where all eigenvalues of the corresponding Jacobian matrix at the FP have strictly negative real parts (thus, FP is stable).

In this framework, the $A_I$-$A_P$-plane is partitioned into different domains of operating regimes (FP-domains). Each FP-domain contains all potential steady states (i.e. FPs) at which the network could operate under the corresponding regime. The FP-domains of operating regimes were determined by using numerical simulations, based on the aforementioned stability criteria obtained analytically. The area of each regime's FP-domain is computed numerically using a sparse grid rule as implemented in Mathematica (version 13).

### Alternative network models

To assess whether, or to what extent, our observed CA1 dynamics in second postnatal week can be explained by other network models (or mechanisms), we created two operationally distinct network models by reparameterizing the STP-RNN model (*Equation 1*). (1) Mono-RNNi ($\theta_P\downarrow$, 'i' for inhibitory GABA): $\theta_p = -0.18$. (2) Mono-RNNe: ($J_I \rightarrow -0.5 \times J_I$, $\theta_P\downarrow$, $\theta_I\downarrow$; 'e' for excitatory GABA): $\theta_p = -0.3$, $\theta_I = -0.1$, $J_I = -1.5$. Either of models (10D) has only one spontaneous FP which is stable (thus, mono-stable) and located at an active state. The properties of these models have been detailed in Results (corresponding text of *Figure 9—figure supplement 1*). According to these parameter values, GABAergic transmission in Mono-RNNe is excitatory rather than inhibitory (note the negative value of its $J_I$). However, it has a weaker excitatory effect than glutamatergic transmission, i.e., $|J_I| < (|J_P| = 6.5)$. Moreover, note that the lower population activity-threshold ($\theta$) of each population reflects that its neurons receive a higher mean level of spontaneous background input and/or have a higher intrinsic excitability (e.g. lower spike-threshold/rheobase or higher membrane resistance, see also *Flossmann et al., 2019*; *Rahmati et al., 2017*).

## Statistical analysis

Statistical analyses were performed using OriginPro 2018 and Microsoft Excel 2010 using the Real Statistics Resource Pack software (Release 7.2, Charles Zaiontz). Several of the presented analyses (e.g. pairwise correlations, PopC, NBs, and motifs) are based on the simultaneous sampling of activity from multiple cells, which renders the FOV our analytical unit. We therefore defined the statistical parameter n as the number of FOVs (dataset related to *Figures 2–5*: P4: 19 FOVs from six mice, P11: 11 FOVs from six mice, P18: 12 FOVs from six mice; dataset related to *Figure 6*: 12 FOVs from six mice), unless otherwise stated. Mouse and FOV IDs are listed in the *Figure 2—source data 1*. All data are reported as mean ± SEM, if not stated otherwise. The Shapiro-Wilk test was used to test for normality. Homogeneity of variances was tested with the Levene's test using the median. For multi-group comparisons, ANOVA was applied for normally distributed data or the Kruskal-Wallis test for non-normally distributed data. In the case of unequal group variances, Welch's correction was applied for the ANOVA. Following a significant result in the ANOVA, post-hoc pairwise comparisons were performed using the Tukey-Kramer (equal variances) or the Games-Howell (unequal variances) test. Following a significant result in the Kruskal-Wallis test, post-hoc pairwise Mann-Whitney U-tests following Holm's approach were performed. p Values (two-tailed tests)<0.05 were considered statistically significant, except for the Shapiro-Wilk test (p<0.01). Details of the statistical tests applied are provided in *Supplementary file 1*.

## Data and code availability

All data analyzed during this study are included in the manuscript, *Supplementary file 1*, and source data files. CATHARSiS is available via GitHub (https://github.com/kirmselab/CATHARSiS).

## Acknowledgements

We thank Ina Ingrisch for excellent technical assistance. This work was supported by Individual Research Grants (KI 1816/6-1, KI 1816/7-1 to KK, HO 2156/5–1, HO 2156/6–1 to KH), the Research Unit 3004 (KI 1816/5-1 to KK, GE 2519/8–1, GE 2519/9–1 to CG), the Priority Program 1665 (HO 2156/3–1/2 to KH, KI 1816/1–1/2 to KK, KI 1638/3–1/2 to SJK), and the CRC Transregio 166 (B2 to CG, B3 to KH, KK) of the German Research Foundation. This publication was supported by the Open Access Publication Fund of the University of Wuerzburg.

## Additional information

### Funding

| Funder | Grant reference number | Author |
| --- | --- | --- |
| Deutsche Forschungsgemeinschaft | KI 1816/1-1/2 | Knut Kirmse |
| Deutsche Forschungsgemeinschaft | HO 2156/3-1/2 | Knut Holthoff |
| Deutsche Forschungsgemeinschaft | GE 2519/8-1 | Christian Geis |
| Deutsche Forschungsgemeinschaft | KI 1638/3-1/2 | Stefan J Kiebel |
| Deutsche Forschungsgemeinschaft | CRC166-B2 | Christian Geis |
| Deutsche Forschungsgemeinschaft | GE 2519/9-1 | Christian Geis |
| Deutsche Forschungsgemeinschaft | CRC166-B3 | Knut Holthoff |
| Deutsche Forschungsgemeinschaft | HO 2156/6-1 | Knut Holthoff |
| Deutsche Forschungsgemeinschaft | HO 2156/5-1 | Knut Holthoff |
| Deutsche Forschungsgemeinschaft | CRC166-B3 | Knut Kirmse |
| Deutsche Forschungsgemeinschaft | KI 1816/7-1 | Knut Kirmse |
| Deutsche Forschungsgemeinschaft | KI 1816/6-1 | Knut Kirmse |
| Deutsche Forschungsgemeinschaft | KI 1816/5-1 | Knut Kirmse |
| University of Wuerzburg | Open Access Publication Fund | Knut Kirmse |

The funders had no role in study design, data collection and interpretation, or the decision to submit the work for publication.

### Author contributions

Jürgen Graf, Formal analysis, Investigation, Methodology, Writing – original draft, Writing – review and editing; Vahid Rahmati, Conceptualization, Formal analysis, Investigation, Methodology, Writing – original draft, Writing – review and editing; Myrtill Majoros, Investigation, Writing – review and editing; Otto W Witte, Christian Geis, Stefan J Kiebel, Supervision, Funding acquisition, Writing – review and editing; Knut Holthoff, Conceptualization, Supervision, Funding acquisition, Writing – review and editing; Knut Kirmse, Conceptualization, Formal analysis, Supervision, Funding acquisition, Investigation, Methodology, Writing – original draft, Writing – review and editing

## Author ORCIDs
Jürgen Graf (iD) http://orcid.org/0000-0001-8160-1016
Vahid Rahmati (iD) http://orcid.org/0000-0002-8969-527X
Christian Geis (iD) http://orcid.org/0000-0002-9859-581X
Stefan J Kiebel (iD) http://orcid.org/0000-0002-5052-1117
Knut Kirmse (iD) http://orcid.org/0000-0002-9206-214X

## Ethics
All animal procedures were performed with approval of the local government (Thüringer Landesamt für Verbraucherschutz, Bad Langensalza, Germany; reference no.: 02-012/16) and complied with European Union norms (Directive 2010/63/EU).

## Decision letter and Author response
Decision letter https://doi.org/10.7554/eLife.82756.sa1
Author response https://doi.org/10.7554/eLife.82756.sa2

---

# Additional files

## Supplementary files
• Supplementary file 1. Statistical tests used in this study. (a) Synopsis of statistical tests related to *Figure 1*. Numerical data are provided in the *Figure 1—source data 1*. (b) Synopsis of statistical tests related to *Figure 2*. Numerical data are provided in the *Figure 2—source data 1*. (c) Synopsis of statistical tests related to *Figure 3*. Numerical data are provided in the *Figure 3—source data 1*. (d) Synopsis of statistical tests related to *Figure 4*. Numerical data are provided in the *Figure 4—source data 1*. (e) Synopsis of statistical tests related to *Figure 5*. Numerical data are provided in the *Figure 5—source data 1*. (f) Synopsis of statistical tests related to *Figure 6* and *Figure 6—figure supplement 2*. Numerical data are provided in the *Figure 6—source data 1*.

• Transparent reporting form

## Data availability
All data analyzed during this study are included in the manuscript, Supplementary File 1 and source data files. CATHARSiS is available via GitHub (https://github.com/kirmselab/CATHARSiS copy archived at swh:1:rev:1524123a889d0fd8ac259b3db64a114f5eb8375e).

The following dataset was generated:

| Author(s) | Year | Dataset title | Dataset URL | Database and Identifier |
|---|---|---|---|---|
| Graf J, Rahmati V, Majoros M, Witte OW, Geis C, Kiebel SJ, Holthoff K, Kirmse K | 2022 | CATHARSiS | https://github.com/kirmselab/CATHARSiS | GitHub, CATHARSiS |

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

## Appendix 1

### Facing a new deadline once the transition to the active state has failed

Here, we address how the network can transition to the active state once a simulated network burst (simNB) failed to converge to the active state (thus, the network returned to the silent state). This can occur in two cases: (1) if the input is relatively strong (see the light green area on the right side of *Figure 7J*) and (2) if the internal deadline is missed (*Figure 8G*). Here, we address this question for the latter case, while our findings will be similarly applicable to the former one.

To this end, we first introduce a third input pulse arriving after the second one whose evoked simNB failed to push the network to the active state (*Figure 8—figure supplement 1A, D*); recall that the first input pulse was used for silencing the network (see *Figure 7C*). We found that in this case, the network encounters a new deadline (*Figure 8—figure supplement 1A, D*; dotted black line #2). In addition, the network expresses a refractory period after the first simNB. Any input (regardless of its strength) arriving during the refractory period will not be able to move the network to the active state and may not be able to trigger a simNB. This results from a weakening of synaptic weights by the first simNB (*Rahmati et al., 2017*), precluding the network from forming the required transient unstable (allowing for simNB emergence) and stable (allowing for transitioning to the active state) fixed points in its fast dynamics. Once the network recovers sufficiently to generate a simNB (*Figure 8—figure supplement 1B, C, E, F*), the countdown for the arrival of a third input to initiate the transition begins (*Figure 8—figure supplement 1G*). As compared to the second input, the third input – with a proper ratio – has a shorter time-window to enable the transition (compare *Figure 8G* and *Figure 8—figure supplement 1G*). This is mainly because of the emerged refractory period. Note that, in contrast to the deadlines, the refractory period is not fixed but has a direct relationship to the size of the preceding simNB; a larger simNB (returning to the rest state) will result in a longer refractory period, which is needed for sufficient synaptic recovery (*Rahmati et al., 2017*). Except for the refractory period, the rest of the mechanisms and responses of the network remain similar to the case of the first deadline (see the corresponding text of *Figure 8*).

Collectively, these results indicate that, in addition to the input ratio, a delicate interaction between the input timing and the network internal dynamics associates with CA1 input-encoding schemes prior to the onset of environmental exploration.

