## [Editor Report]

This study provides fundamental findings about the developing brain and compelling evidence for how hippocampal physiology evolves during the first few postnatal weeks. Unlike previous in vitro results, which find declining network synchrony after the first postnatal week, the authors find in vivo that synchrony increases and peaks in the second postnatal week, despite emerging GABA-mediated inhibition during this time. They develop a model to explain these findings and suggest an underlying bistable population dynamic, oscillating between silent and active states, that sculpts input discrimination and network synchrony.

---

## [Decision Letter]

**Decision letter after peer review:**

[Editors’ note: the authors submitted for reconsideration following the decision after peer review. What follows is the decision letter after the first round of review.]

Thank you for submitting the paper "Network instability dynamics drive a transient bursting period in the developing hippocampus in vivo" for consideration by *eLife*. Your article has been reviewed by 3 peer reviewers, and the evaluation has been overseen by a Reviewing Editor and a Senior Editor. The reviewers have opted to remain anonymous.

Comments to the Authors:

We are sorry to say that, after consultation with the reviewers, we have decided that this work will not be considered further for publication by *eLife*. Given the reviewers' enthusiasm of the manuscript, if you feel you can address the reviewers' concerns with additional data collection and analyses, we welcome submission of a revised manuscript, should you choose to decide to collect new data. You can refer to this manuscript number, but we cannot make any guarantees about acceptance because the work would be reconsidered as a new submission.

All reviewers expressed enthusiasm for this manuscript and thought the results were of broad interest to the readership of *eLife*. While individual reviews are included below, here are a couple of points where reviewers agreed were essential to be included in a revised version before consideration for publication at *eLife*.

1) Anesthesia has a major effect on neuronal dynamics and therefore, it might seriously impact the findings of the study. The authors should provide experimental data from non-anesthetized animals to confirm their results. This will augment the relevance and validity of the study.

2) A second major aspect that needs to be carefully addressed in a revised version is the data analysis and modelling limitations. All three reviewers raised this aspect. Please see their detailed comments and suggestions below.

*Reviewer #1 (Recommendations for the authors):*

This interesting manuscript by Graf and colleagues aims to map the developmental trajectories of spontaneous network activity of the developing hippocampus. The authors perform in vivo calcium imaging of CA1 neurons throughout development at P4, P11, and P18. They first develop a computational pipeline to accurately extract neural sources and assign timeseries GCaMP fluorescence values, which is challenging in dense and overlapping cell populations. They then identify that network synchrony (which the authors equate with network burstiness) peaks in the second postnatal week. They found this unexpected because prior in vitro results have identified network synchrony primarily in the first postnatal week, and emerging GABA inhibition is thought to gradually reduce network synchrony thereafter. Using a recurrent neural network model, assuming a simple recurrent architecture within and between excitatory and inhibitory neurons, the authors identify bistable regimes, that amplify input in different and non-linear ways. Silent states were found to amplify input that leads to burstiness, whereas active states did not lead to bursting network behaviors. In sum, the authors propose that bistable network properties in the second week of postnatal life may be important for generating synchronous network activity and performing input discrimination prior to environmental exploration and experience-dependent learning.

The strengths of this study are the systematic characterization of spontaneous CA1 network activity, which was done in vivo, and longitudinally, across the first three postnatal weeks. Rigor was taken to collect high quality data in a challenging prep and the combination of experiments and modeling led to the proposal of an interesting model involving bistable dynamics that may be broadly relevant to developmental physiology. The observation that burstiness is due to single neurons having higher coupling with population activity , not due to increased pairwise correlations, was also quite interesting. Overall the claims of the study are justified by their data.

A main weakness or concern is that 1. it is not clear how functionally important p11 synchrony/burstiness is, and 2. while one network mechanism is proposed there may be other underlying network dynamics that can explain p11 burstiness equally well or better. For instance, it's possible that emerging GABA-ergic inhibition acts on other interneurons or on highly patterned set of principle neurons, or that the sub threshold properties of principle neurons change dramatically during this P11 window, such that any of these alternative mechanisms may drive the observed bursting behavior. Further explorations of the model, to negate alternative explanations, or experimental perturbations during P4 vs P11 vs P18, would clarify and strengthen the main conclusions.

Other Points:

1. Figures 2 and 3 rely heavily on CDFs but a plain display of histograms would be more informative and it would be easier to evaluate heavy tail vs normal distributions, say of firing rates.

2. The enhanced burstiness on P11 seems very sensitive to the definition used for burstiness (ie NB). For instance fraction of time (Figure 2D) suggests similar burstiness on days p11 and p18, whereas burstiness duration (Figure 2E) suggests similar levels on P4 an P11, thus it is not clear how robust or important the p11 bursting behavior is.

3. In Figure 4, coupling to population activity, and all Pearsons analyses, should control for increases in overall firing rates after P4.

4. The model being used seems to be an extension of prior models that are well validated with existing experimentally determined constraints, but such validation data should again be shown for this new extended model.

To strengthen the claim that P11 burstiness is functionally important it would be useful to perform in silico manipulations, or actual experimental manipulations, possibly silencing of these P11 bursts, to show functional consequences later in development.

To strengthen the claim that underlying network bistabiliy leads to this burstiness, it would be useful to provide in silico manipulations that support this, or test alternative models to show they do not lead to burstiness.

The enhanced burstiness on P11 seems very sensitive to how burstiness is defined. It may be important to perform these analyses using a wide range of definitions to show the results are robust to small changes in definition.

In general, data presentation rely heavily on CDFs, but it would be easier to interpret and evaluate if histograms of the raw data were provided (ie for Figures 2 and 3).

All Pearsons analyses, should control for increases in overall firing rates after P4, by shuffling the datasets and providing chance calculations.

More validation data for the model would build confidence in the modeling results.

Overall, the manuscript was difficult to read , possibly because certain terms are used interchangeably (synchrony and burstiness) and possibly because enough of the methods are not described in the main text and possibly because the writing sometimes meanders and loses a consistent message. A tightening up of the text would be very helpful.

*Reviewer #2 (Recommendations for the authors):*

Strengths:

The paper is very careful to extract single cell signals from the densely populated CA1 region and uses a number of appropriate analysis methods to quantify single cell and population dynamics. Their analysis approaches allowed them to determine differences at distinct developmental stages that could have easily been missed. Their detection methods and barrage of analysis methods will be generally useful to any field that studies functional calcium signals at the network level.

The paper nicely combines experimental findings with computational modeling to gain insight into the development of a functional dorsal CA1. Their experimental findings are on face value difficult to reconcile, but their computational modeling work brings together their experimental findings, along with those from other papers, to put forward a comprehensive framework. This paper is a good example of how experiments should inform computational models to bring insight into brain function.

Weaknesses:

Animals are not awake/alert during imaging. They have just undergone surgery (60 mins prior to imaging) and are in a sedative state during imaging (as far as I can tell). This is a major weakness of the paper as no doubt the CA1 will behave differently in an awake state. This makes it difficult to generalize their findings to the awake state.

The paper contains a lack of causal relationships. For instance, do the NBs setup the hippocampus for learning right before environmental exploration, or do they have some other role? Could they be epiphenomenal? There are no experimental manipulations of NBs, which are needed to further test the authors theories.

A big part of the proposed mechanism for increased NBs at P11 is that GABA has switched from excitatory to inhibitory at all synapses in CA1 by P11, but is this true? The authors refer to literature, but do not show that GABA is inhibitory in their experiments at this time point in CA1.

There is a lack of explanation of why P4 networks have more in common with P18 networks than P11 networks, in many cases. The data clearly demonstrates that there is not a progression of network dynamics as a function of age, and instead P11 is, for many measures, behaving differently than both younger and older developmental stages.

Analysis of FOVs is performed on separate animals at the different ages. The findings would be further strengthened if the same neurons were tracked over time. This can be done in adult mice. However, given the developmental changes that occur between P4 and P18, this type of experiment may not be feasible with current methods. Still, it would be insightful to observe how the same network develops throughout this period. An experiment for the future, perhaps.

It is not explicitly clear whether the mice are awake during the experiments or under anesthesia. It is stated that head-fixed animals are "spontaneously breathing" in the Results section, and in the Methods they state "Isoflurane was discontinued after completion of the surgical preparation and gradually substituted with the analgesic sedative nitrous oxide". So, what is the general state of the animals during in vivo imaging? This is important as it will certainly affect network activity in CA1 and should be discussed.

The data is interpreted that NBs are more prevalent at P11 than P4 and P18. However, given the overall increase in CAT frequency at P18 (Figure 3A, right) it might make it more difficult to detect isolated NBs from those riding on top of (or very close in time to) other NBs (especially given that calcium transients have relatively slow decay kinetics). The authors should be careful to make sure their NB detection method is not biasing them to detect more NBs in FOVs with generally lower activity, i.e. at P11 versus P18.

I do wonder if CAT kinetics differ in PCs at the different age groups. They should show that expression levels are similar, rise times/decay times are similar, noise levels are similar…to rule out these as confounds to the other forms of analysis.

*Reviewer #3 (Recommendations for the authors):*

The manuscript addresses an important topic. The data and modeling results provide new insights into the developmental trajectories of network activity in hippocampal CA1 area. However, several major aspects, especially concerning (i) the solidity of data from a rather low number of mice, (ii) the lack of experimental data uncovering the underlying mechanisms of described processes and (iii) the interpretation of results in the context of existing literature, dampen my enthusiasm and need to be addressed as part of a major revision before further consideration of the study.

1. The in vivo dataset is too small to enable reliable conclusions. In the manuscript, the number of mice used for each analysis is not specified. In general, n numbers should be stated more clearly and be included in the figure legends. For each mouse 3-5 FOVs and in average 14 FOVs/group were acquired, implying that only around 3-4 mice were used for group analysis. Furthermore, the applied stats use FOVs as statistical unit and covers not for single datapoint independency, i.e. FOVs that are coming from the same mouse. The authors might think about the use of mixed-effect models for statistical analysis after increasing the size of the dataset. Given the small size of the recording area, the authors should state how overlapping FOVs and thus, cell populations, between imaging sessions were avoided. Moreover, what was the rationale for focusing the investigation on P4, P11, and P18? Are these time points of particular relevance? In the absence of more time points, it is unclear how the dynamics of described processes evolve.

2. Besides modeling, additional experimental evidence of the cellular interactions underlying the developmental dynamics of bursts should be added. Direct targeting of distinct neuronal populations, their acute or chronic manipulation, possible combination with electrophysiological recordings, are just few suggestions, how the insights from modeling should be complemented. Moreover, the RNN model provides insights into the mechanisms governing the elevated burstiness constrained to the P11 age group. However, it remains unclear, which mechanisms potentially contribute to the developmental emergence of a bi-stable network as well as its potential disappearance. The authors might include this developmental aspect in the model as well and discuss age-dependent features in synaptic strength and timing that account for observed changes in network synchrony and burstiness in more detail.

3. line 371-374: the authors conclude the presence of a lower synchrony in the developing CA1 area compared to sensory cortices as the result of the identification of lower correlation values. The reference cited for visual cortex (Rochefort at al., 2009) uses no pairwise correlation analysis and should be removed. The other references for somatosensory cortices quantify pairwise correlation but use different analytical strategies and not STTC as used in the present manuscript. Thus, the comparison of absolute correlation values might be inappropriate and further depend on the chosen timescale. Another important factor impacting correlation values is the use of anesthesia. Mice were anesthetized with nitrous oxide, known to alter neurotransmission and consequently affecting physiological activity. While anesthesia increases correlation in sensory areas (Goltstein et al., 2015), it decreases STTC values in the CA1 area (Yang et al., 2021). The studies cited in line 371-374 are done in non-anesthetized or in urethane/isoflurane anesthetized mice. Consequently, the identified lower correlation values could thus be an artifact of differential actions of anesthesia in sensory areas versus CA1 area. Moreover, anesthesia has been identified to impact brain activity in an age- and dose-dependent manner (Chini et al., 2019) and might therefore impact the selected age groups differently. Accordingly, the authors should refrain from the statement of a developmental lower correlation in CA1 area than in sensory areas. To support this statement additional recordings in non-anesthetized mice in CA1 area and compared to equally analyzed open-access datasets of sensory cortices would be required. In line with this, PSD analysis of frequencies below 1 Hz as used in Figure 3B are in particular sensitive to anesthesia and should be interpreted with caution.

4. The threshold for burst detection was quantified individually for each FOV. Consequently, the proportion of silent periods in each FOV affects the threshold calculation and might affect the P11 age group, where activity is almost but not completely continuous, differently. Analysis of bursts detected with a threshold quantified only on "active" periods by excluding silent periods could further support the presence of burst activity during events without be affected by the changing discontinuity over age.

5. The authors describe in the Introduction that in the neonatal hippocampus activity is triggered by myoclonic twitches. Did the authors monitor the animal's movement? If yes, the occurrence of bursts and presence of network motifs could be correlated to the movement. This would enable a better understanding of how age-dependent dissociation of CA1 activity from twitches relates to the emergence and stabilization of self-organized hippocampal motifs during recurring activation patterns (line 258, 259) and changes in neuronal synchrony.

[Editors’ note: further revisions were suggested prior to acceptance, as described below.]

Thank you for resubmitting your work entitled "Network instability dynamics drive a transient bursting period in the developing hippocampus in vivo" for further consideration by *eLife*. Your revised article has been evaluated by Laura Colgin (Senior Editor) and a Reviewing Editor.

The manuscript has been improved but there are some remaining issues that need to be addressed, as outlined below:

While all the reviewers remain enthusiastic about this work, Reviewer 3 points out some major concerns that need to be addressed before this manuscript can be recommended for publication.

I appreciate the work that was done to investigate the influence that N20 anesthesia has on hippocampal network activity, but I find that the conclusions that the authors draw from these experiments is not rooted in the data. Furthermore, and perhaps more concerning, the network activity of P11 "anesthetized" mice in this dataset seemingly contradict the main results that are presented in the remainder of the manuscript.

1) The activity of anesthetized P11 mice displayed in figure 6 greatly differs from anesthetized age-matched mice in the rest of the manuscript.

2) The average theta bandpower of N20 mice in figure S3D is less than half of those of age-matched mice in figure 3E (~3.7x10-4 vs 8.9x10-4). These values would be much more similar to P4 (2.9x10-4) and P18 (3.4x10-4) mice than to the P11 ones (8.9x10-4) that are plotted in figure 3E.

3) The peak in the Φ PSD of figure 6F for both anesthetized and unanesthetized mice is virtually indistinguishable to that of P18 mice in figure 3D (and roughly half the size of that of P11 mice).

4) The N20 mice in figure S3G have a % of cells with a significant STTC that is less than half than age-matched mice in figure 4E. The values in figure S3G are actually much closer to the P18 group than the P4 one. How can that be?

5) Could the authors verify whether anesthesia affects other parameters that are central to the thesis of the manuscript such as the Gini coefficient of CaT etc.?

6) I find the provided data does not corroborate the statement that neuronal network activity under nitrous oxide closely resembles that recorded in unanesthetized mice. Several parameters that are important to the thesis of the manuscript such as STTC, population coupling etc. are affected by anesthesia. As a general concern, providing a comparison between anesthetized and unanesthetized mice at one single developmental timepoint does not give much insight into whether the developmental processes that are described in the paper are biased by anesthesia. While experimentally addressing this concern is time consuming, it should be discussed.

7) Line 125: what is this event detection routine based on analyzing mean ∆F(t) that you compared CATHARSIS to? Overall, the information provided to assess the quality of the calcium transient extraction pipeline is scarce, and its validity has to be taken at face value. It would be comparable to an electrophysiology paper using its unique spike sorting algorithm. While I can understand that extracting calcium transients from a densely packed brain area such as the rodent CA1 has its own unique challenges, this is now routinely done using established pipelines. For instance, a recent paper (Dard et al., 2022, also published in *eLife*) used suite2p to extract calcium transients from the developing CA1 network (same developmental phase). I would suggest the authors to attempt at replicating their results using this more established calcium transient pipeline.

8) Line 189: figure 3C and the manner in which the term "continuous activity" is used in this paragraph is perplexing. Brain activity is discontinuous (alternation of low-activity (silent) state and high activity periods) in early development, but it should already be continuous and adult-like at P18 if not already at P11. It is therefore perplexing that several P18 mice have a time in continuous activity (Figure 2C) that is well below 50% and a few even below 20%. If this is an effect of anesthesia, it is concerning and a datapoint that goes in the direction of N2O having a major impact on hippocampal activity (anesthesia seems indeed to reduce CaT frequency also in the data presented here, see Figure S3C). If this depends on the manner in which "continuous activity" is defined/computed, perhaps a different term should be used.

9) The authors write that a statistical mixed-effect model is not applicable in their opinion due to low numbers of FOVs/mouse. However, the data are not independent and a mouse with 4 FOVs biases the data distributions much stronger than a mouse with only 1 FOV. This is especially important since an overlap of FOVs cannot be ruled out as they write in their methods. If the authors don't want to use mixed-effect models, they should take mice as statistical unit by taking the average of each mouse, as it's also done in many of their cited studies. A minor point here, in Figure 4 D the number of neurons is written in the legends. For P11 mice 11 FOVs and for P18 mice 12 FOVs were recorded, resulting in 161 neurons for P11 and 100 neurons for P18, respectively. How do the authors explain the substantially higher yield in P11 mice? Might differential effects of anesthesia play a role?

10) I appreciate that, in the revised manuscript, the authors considered the concerns regarding the unwanted effects of anesthesia and performed additional experiments in unanesthetized P11 mice. However, as they write, anesthesia has age-dependent effects on network activity. Especially, the emergence of an active sleep-wake cycle (at ~P14) is suggested to co-occur with frequency-specific (i.e. altered network dynamics) effects of anesthesia (Ackman et al., 2014; Chini et al., 2019; Cirelli and Tononi, 2015; Shen and Colonnese, 2016). Thus, their anesthetic strategy might affect their comparisons between P18 and younger mice.

*Reviewer #1 (Recommendations for the authors):*

The authors addressed all my concerns.

*Reviewer #2 (Recommendations for the authors):*

The authors have added new experiments in non-anesthetized mice, and substantial new analysis, modeling, and interpretation. Through this revision they have addressed all of my previous concerns.

*Reviewer #3 (Recommendations for the authors):*

The manuscript by Graf et al., investigates the population dynamics of the mouse CA1 hippocampal area across the first two weeks of extra-uterine life. The manuscript leverages the combination of experimental data and modeling to identify P11 as a developmental stage in which network burstiness peaks due to network bi-stability. An increase in synaptic inhibition is suggested as being the reason behind this transient network characteristic.

The revised manuscript employs a series of innovative and state-of-the-art analytical techniques and the modeling work is elegantly used to try to get mechanistic insight into the processes underlying the network properties and how they evolve throughout development. The modeling section of the paper is very high quality, yet it is not always clear how it relates (or how it explains) the experimental data that is presented in the first part of the manuscript (see below for comments). The authors try to explain the link between experimental and modelling work by discussing published data from in vitro, electrophysiological and imaging studies in CA1 but also in sensory areas. Due to the methodological variability as well as diverse selected age ranges in these studies, it is unclear how they relate to the ones included in the manuscript. Although the authors might refrain from doing manipulation experiments, they could image GABAergic neurons to better understand their contributions to NBs.

In line with the concerns listed below, the manuscript does provide solid experimental and theoretical evidence for its conclusions.

I appreciate the work that was done to investigate the influence that N20 anesthesia has on hippocampal network activity, but I find that the conclusions that the authors draw from these experiments is not rooted in the data. Furthermore, and perhaps more concerning, the network activity of P11 "anesthetized" mice in this dataset seemingly contradict the main results that are presented in the remainder of the manuscript.

– The activity of anesthetized P11 mice displayed in figure 6 greatly differs from anesthetized age-matched mice in the rest of the manuscript.

– The average theta bandpower of N20 mice in figure S3D is less than half of those of age-matched mice in figure 3E (~3.7x10-4 vs 8.9x10-4). These values would be much more similar to P4 (2.9x10-4) and P18 (3.4x10-4) mice than to the P11 ones (8.9x10-4) that are plotted in figure 3E.

– The peak in the Φ PSD of figure 6F for both anesthetized and unanesthetized mice is virtually indistinguishable to that of P18 mice in figure 3D (and roughly half the size of that of P11 mice).

– The N20 mice in figure S3G have a % of cells with a significant STTC that is less than half than age-matched mice in figure 4E. The values in figure S3G are actually much closer to the P18 group than the P4 one. How can that be?

– Could the authors verify whether anesthesia affects other parameters that are central to the thesis of the manuscript such as the Gini coefficient of CaT etc.?

– I find the provided data does not corroborate the statement that neuronal network activity under nitrous oxide closely resembles that recorded in unanesthetized mice. Several parameters that are important to the thesis of the manuscript such as STTC, population coupling etc. are affected by anesthesia.

As a general concern, providing a comparison between anesthetized and unanesthetized mice at one single developmental timepoint does not give much insight into whether the developmental processes that are described in the paper are biased by anesthesia. While experimentally addressing this concern is time consuming, it should be discussed.

Line 125: what is this event detection routine based on analyzing mean ∆F(t) that you compared CATHARSIS to? Overall, the information provided to assess the quality of the calcium transient extraction pipeline is scarce, and its validity has to be taken at face value. It would be comparable to an electrophysiology paper using its unique spike sorting algorithm. While I can understand that extracting calcium transients from a densely packed brain area such as the rodent CA1 has its own unique challenges, this is now routinely done using established pipelines. For instance, a recent paper (Dard et al., 2022, also published in *eLife*) used suite2p to extract calcium transients from the developing CA1 network (same developmental phase). I would suggest the authors to attempt at replicating their results using this more established calcium transient pipeline.

Line 189: figure 3C and the manner in which the term "continuous activity" is used in this paragraph is perplexing. Brain activity is discontinuous (alternation of low-activity (silent) state and high activity periods) in early development, but it should already be continuous and adult-like at P18 if not already at P11. It is therefore perplexing that several P18 mice have a time in continuous activity (Figure 2C) that is well below 50% and a few even below 20%. If this is an effect of anesthesia, it is concerning and a datapoint that goes in the direction of N2O having a major impact on hippocampal activity (anesthesia seems indeed to reduce CaT frequency also in the data presented here, see Figure S3C). If this depends on the manner in which "continuous activity" is defined/computed, perhaps a different term should be used.

The authors write that a statistical mixed-effect model is not applicable in their opinion due to low numbers of FOVs/mouse. However, the data are not independent and a mouse with 4 FOVs biases the data distributions much stronger than a mouse with only 1 FOV. This is especially important since an overlap of FOVs cannot be ruled out as they write in their methods. If the authors don't want to use mixed-effect models, they should take mice as statistical unit by taking the average of each mouse, as it's also done in many of their cited studies. A minor point here, in Figure 4 D the number of neurons is written in the legends. For P11 mice 11 FOVs and for P18 mice 12 FOVs were recorded, resulting in 161 neurons for P11 and 100 neurons for P18, respectively. How do the authors explain the substantially higher yield in P11 mice? Might differential effects of anesthesia play a role?

I appreciate that, in the revised manuscript, the authors considered the concerns regarding the unwanted effects of anesthesia and performed additional experiments in unanesthetized P11 mice. However, as they write, anesthesia has age-dependent effects on network activity. Especially, the emergence of an active sleep-wake cycle (at ~P14) is suggested to co-occur with frequency-specific (i.e. altered network dynamics) effects of anesthesia (Ackman et al., 2014; Chini et al., 2019; Cirelli and Tononi, 2015; Shen and Colonnese, 2016). Thus, their anesthetic strategy might affect their comparisons between P18 and younger mice.

---

## [Author Response]

[Editors’ note: the authors resubmitted a revised version of the paper for consideration. What follows is the authors’ response to the first round of review.]

Comments to the Authors:All reviewers expressed enthusiasm for this manuscript and thought the results were of broad interest to the readership of eLife. While individual reviews are included below, here are a couple of points where reviewers agreed were essential to be included in a revised version before consideration for publication at eLife.

We thank the reviewers for considering our results of broad interest to the readership of *eLife*. We thoroughly addressed all points raised by the reviewers and extensively dealt with the reviewers’ points of critique, especially those related to (1) anesthesia and (2) analysis/modeling as detailed below.

1) Anesthesia has a major effect on neuronal dynamics and therefore, it might seriously impact the findings of the study. The authors should provide experimental data from non-anesthetized animals to confirm their results. This will augment the relevance and validity of the study.

We now provide an additional dataset in which we compare network activity measured under sedation/anesthesia with nitrous oxide with that in unanesthetized mice using a paired experimental design. Our analyses confirm major previous conclusions and further provide minor quantitative differences between the two conditions. We thank the reviewers for this suggestion, as the new dataset clearly augments the relevance and validity of our study.

2) A second major aspect that needs to be carefully addressed in a revised version is the data analysis and modelling limitations. All three reviewers raised this aspect. Please see their detailed comments and suggestions below.

We substantially extended both the data analysis and network modeling sections. For details, please see our point-to-point reply below. We thank the reviewers for their valuable suggestions that helped to substantiate our conclusions.

Changes to our manuscript include:

1. We added a novel dataset addressing the effects of nitrous oxide on vital parameters, body movements and neuronal dynamics at P11 (new Figure 6, new Figure S2, new Figure S3).

2. We included an analysis of the functional significance of the STP-RNN network model, particularly in relation to developmental changes in accessibility of different operating regimes (ISN, non-ISN) and network burstiness (new Figure 9).

3. We extended the computational modeling to explore whether alternative network models (or mechanisms) are better suited to explain the experimentally measured neuronal dynamics (new Figure S5).

4. We confirmed the robustness of the reported developmental changes in network burstiness by investigating a wide range of NB definitions (new Figure S1).

5. We quantified (dis-)continuity of network activity throughout development (extended Figure 3A–C).

6. We made CATHARSiS available via GitHub (https://github.com/kirmselab/CATHARSiS). To simplify reuse by others, we provide (i) a detailed user manual, (ii) code to generate demo files that can be used for testing purposes, (iii) a GUI for template selection and (iv) a Matlab app for the visual exploration of detection results.

Reviewer #1 (Recommendations for the authors):This interesting manuscript by Graf and colleagues aims to map the developmental trajectories of spontaneous network activity of the developing hippocampus. The authors perform in vivo calcium imaging of CA1 neurons throughout development at P4, P11, and P18. They first develop a computational pipeline to accurately extract neural sources and assign timeseries GCaMP fluorescence values, which is challenging in dense and overlapping cell populations. They then identify that network synchrony (which the authors equate with network burstiness) peaks in the second postnatal week. They found this unexpected because prior in vitro results have identified network synchrony primarily in the first postnatal week, and emerging GABA inhibition is thought to gradually reduce network synchrony thereafter. Using a recurrent neural network model, assuming a simple recurrent architecture within and between excitatory and inhibitory neurons, the authors identify bistable regimes, that amplify input in different and non-linear ways. Silent states were found to amplify input that leads to burstiness, whereas active states did not lead to bursting network behaviors. In sum, the authors propose that bistable network properties in the second week of postnatal life may be important for generating synchronous network activity and performing input discrimination prior to environmental exploration and experience-dependent learning.

We highly appreciate the reviewer’s detailed evaluation of our study.

The strengths of this study are the systematic characterization of spontaneous CA1 network activity, which was done in vivo, and longitudinally, across the first three postnatal weeks. Rigor was taken to collect high quality data in a challenging prep and the combination of experiments and modeling led to the proposal of an interesting model involving bistable dynamics that may be broadly relevant to developmental physiology. The observation that burstiness is due to single neurons having higher coupling with population activity , not due to increased pairwise correlations, was also quite interesting. Overall the claims of the study are justified by their data.

We are pleased that the reviewer considers our conclusions to be overall justified based on the data and analyses provided.

A main weakness or concern is that 1. it is not clear how functionally important p11 synchrony/burstiness is, and 2. while one network mechanism is proposed there may be other underlying network dynamics that can explain p11 burstiness equally well or better. For instance, it's possible that emerging GABA-ergic inhibition acts on other interneurons or on highly patterned set of principle neurons, or that the sub threshold properties of principle neurons change dramatically during this P11 window, such that any of these alternative mechanisms may drive the observed bursting behavior. Further explorations of the model, to negate alternative explanations, or experimental perturbations during P4 vs P11 vs P18, would clarify and strengthen the main conclusions.

In the revised manuscript, we address the two main concerns of the reviewer in detail. (1) In the model, we demonstrate that the developmental emergence of bistability is accompanied by an effective availability of inhibition-stabilized states at P11. The functional importance of these changes for hippocampal development is analyzed and discussed. (2) We explore alternative network mechanisms underlying burstiness at P11. We find that bistability in the presence of inhibitory GABA robustly explains our experimental observations. We are grateful for the reviewer’s suggestions as the novel results clearly strengthen our main line of argumentation.

For a detailed response, please see the “Reviewer #1 (Recommendations for the authors)” section below.

Other Points:1. Figures 2 and 3 rely heavily on CDFs but a plain display of histograms would be more informative and it would be easier to evaluate heavy tail vs normal distributions, say of firing rates.

Please see below.

2. The enhanced burstiness on P11 seems very sensitive to the definition used for burstiness (ie NB). For instance fraction of time (Fig 2D) suggests similar burstiness on days p11 and p18, whereas burstiness duration (Fig 2E) suggests similar levels on P4 an P11, thus it is not clear how robust or important the p11 bursting behavior is.

Please see below.

3. In Figure 4, coupling to population activity, and all Pearsons analyses, should control for increases in overall firing rates after P4.

Please see below.

4. The model being used seems to be an extension of prior models that are well validated with existing experimentally determined constraints, but such validation data should again be shown for this new extended model.

Please see below.

To strengthen the claim that P11 burstiness is functionally important it would be useful to perform in silico manipulations, or actual experimental manipulations, possibly silencing of these P11 bursts, to show functional consequences later in development.To strengthen the claim that underlying network bistabiliy leads to this burstiness, it would be useful to provide in silico manipulations that support this, or test alternative models to show they do not lead to burstiness.

Our data indicate that burstiness at P11 can be seen as an expression of the complex dynamics in a bi-stable STP-RNN (we now explicit this point in line 459). Bi-stability in the model is reminiscent of our experimental observations demonstrating that CA1 dynamically transitions between discontinuous and continuous activity states at P11 (new Figure 3A–C). Importantly, we further demonstrate that the developmental emergence of bistability is accompanied by an effective availability of inhibition-stabilized network (ISN) regimes in the second postnatal week. As ISN regimes have been linked to sparse and efficient information processing in the adult brain, their emergence in the second postnatal week probably reflects an important milestone in circuit development. We added the new Figure 9 and new Figure S5 and extended the Results sections (lines 438–461) to present these novel findings and their functional implications during CA1 development.

We emphasize, however, that the (causative) developmental functions of CA1 network bursts remain hypothetical at present. In line with the editor’s recommendation regarding additional experiments, we feel that analyzing their developmental effects through actual experimental perturbations is far beyond the scope of the present manuscript.

To strengthen the claim that underlying network bistabiliy leads to this burstiness, it would be useful to provide in silico manipulations that support this, or test alternative models to show they do not lead to burstiness.

We substantially extended our computational modeling to explore whether alternative network models (or mechanisms) are better suited to explain the observed neuronal dynamics (new Figure S5). Specifically, we considered two alternative mono-stable network models with either inhibitory or excitatory GABA for comparison. Our data indicate that a bi-stable STP-RNN model with inhibitory GABA best explains the experimental observations. We added lines 401– 436 to Results.

The enhanced burstiness on P11 seems very sensitive to how burstiness is defined. It may be important to perform these analyses using a wide range of definitions to show the results are robust to small changes in definition.

We thank the reviewer for his/her constructive suggestion. To address this point, we systematically varied our operational definition of NBs in two ways. (1) First, we systematically varied the integration window Δt that is used to compute an activity-dependent threshold for NB detection, separately for each field of view (new Figure S1A). (2) Second, we used a constant (i.e. activity-independent) threshold, which we applied to all fields of view from the three age groups (new Figure S1B). Our results confirm that the reported developmental changes in network burstiness are robust to a wide range of NB definitions. We added the new Figure S1, added a sentence to Results (204–206) and extended the “network bursts” section in Methods (1104–1110).

In general, data presentation rely heavily on CDFs, but it would be easier to interpret and evaluate if histograms of the raw data were provided (ie for Figures 2 and 3).

Thank you for the suggestion. We replaced CDFs by standard histograms in Figure 2B (CaT frequency) and Figure 3I (Participation rate). We also used histograms in the new Figure 3B (Active cells). We prefer keeping the CDF in Figure 2F (CV2), as the age-dependent changes were less obvious in standard histograms.

All Pearsons analyses, should control for increases in overall firing rates after P4, by shuffling the datasets and providing chance calculations.

We already considered this important point in the initial manuscript. We assessed the significance of population coupling on the basis of surrogate data generated by shuffling CaT onset times (see Methods, lines 1137–1152, and Figure 4A–C) and, thus, controlling for differences in CaT rates. Likewise, for quantification of pairwise correlations, we used the spike time tiling coefficient (STTC), which is inherently insensitive to event frequency (Cutts et al., 2014). In addition, the significance of STTC values was examined on the basis of surrogate data generated by shuffling CaT onset times (see Methods, lines 1129-1135, and Figure 4E-G).

More validation data for the model would build confidence in the modeling results.

The reviewer correctly mentioned that the STP-RNN model builds on our previous work (Flossmann et al., 2019; Rahmati et al., 2017). Important new constraints of the model used in this manuscript are: (1) We consider GABA to be inhibitory at P11. This constraint is based on recent studies indicating that GABAergic interneurons exert (net) synaptic inhibition already by the end of the first postnatal week (Murata et al., 2020; Valeeva et al., 2016). (2) Our experimental data indicate that, at P11, CA1 networks transition between discontinuous and continuous activity states. Here, “discontinuous” implies that the network spends considerable time in a silent state (from which NBs may emerge or not), whereas “continuous” activity refers to sustained non-zero network activity. We quantified the degree of continuity for the age groups (new Figure 3C), added the distributions of active cells per frame (new Figure 3B) and extended Figure 3A for a more comprehensive illustration. In the modeling part, we refer to these experimental constraints in lines 299–307. We also emphasize in Results that the model behavior is in agreement with our experimental data indicating that NBs are chiefly generated during periods of discontinuous activity, i.e. they emerge from the silent (see Figure 3A–C).

As mentioned above, we substantially extended our computational modeling to explore whether alternative network models are better suited to explain the observed neuronal dynamics (new Figure S5, new Figure 9, lines 401–461 in Results). Our data indicate that a bi-stable STP-RNN model with inhibitory GABA robustly explains the experimental observations.

Overall, the manuscript was difficult to read , possibly because certain terms are used interchangeably (synchrony and burstiness) and possibly because enough of the methods are not described in the main text and possibly because the writing sometimes meanders and loses a consistent message. A tightening up of the text would be very helpful.

We carefully edited the entire manuscript to further improve readability and clarity.

In this context, we also replaced the term “synchrony” by “network burstiness” or “NBs” throughout the manuscript.

Reviewer #2 (Recommendations for the authors):Strengths:The paper is very careful to extract single cell signals from the densely populated CA1 region and uses a number of appropriate analysis methods to quantify single cell and population dynamics. Their analysis approaches allowed them to determine differences at distinct developmental stages that could have easily been missed. Their detection methods and barrage of analysis methods will be generally useful to any field that studies functional calcium signals at the network level.The paper nicely combines experimental findings with computational modeling to gain insight into the development of a functional dorsal CA1. Their experimental findings are on face value difficult to reconcile, but their computational modeling work brings together their experimental findings, along with those from other papers, to put forward a comprehensive framework. This paper is a good example of how experiments should inform computational models to bring insight into brain function.

We appreciate the reviewer’s detailed evaluation of our study and are pleased about his/her positive feedback.

Weaknesses:Animals are not awake/alert during imaging. They have just undergone surgery (60 mins prior to imaging) and are in a sedative state during imaging (as far as I can tell). This is a major weakness of the paper as no doubt the CA1 will behave differently in an awake state. This makes it difficult to generalize their findings to the awake state.

We performed additional set of experiments in which we compare network activity measured under nitrous oxide with that in unanesthetized mice using a paired experimental design. Our analyses confirm major previous conclusions and further provide minor quantitative differences between the two conditions. We thank the reviewers for this suggestion, as the new dataset clearly augments the relevance and validity of our study.

For a detailed response, please see the below.

The paper contains a lack of causal relationships. For instance, do the NBs setup the hippocampus for learning right before environmental exploration, or do they have some other role? Could they be epiphenomenal? There are no experimental manipulations of NBs, which are needed to further test the authors theories.

Our data indicate that burstiness at P11 can be seen as an expression of the complex dynamics in a bi-stable STP-RNN (we now explicit this point in line 459). Bi-stability in the model is reminiscent of our experimental observations demonstrating that CA1 dynamically transitions between discontinuous and continuous activity states at P11 (new Figure 3A–C). Importantly, we now demonstrate that the developmental emergence of bistability is accompanied by an effective availability of inhibition-stabilized network (ISN) regimes in the second postnatal week. As ISN regimes have been linked to sparse and efficient information processing in the adult brain, their emergence in the second postnatal week probably reflects an important milestone in circuit development. We added the new Figure 9 and new Figure S5 and extended the Results sections (lines 438–461) to present these novel findings and their functional implications during CA1 development.

We emphasize, however, that the (causative) developmental functions of CA1 network bursts remain hypothetical at present. In line with the editor’s recommendation regarding additional experiments, we feel that analyzing their developmental effects through actual experimental perturbations is far beyond the scope of the present manuscript.

A big part of the proposed mechanism for increased NBs at P11 is that GABA has switched from excitatory to inhibitory at all synapses in CA1 by P11, but is this true? The authors refer to literature, but do not show that GABA is inhibitory in their experiments at this time point in CA1.

The available in vivo evidence (CA1) clearly indicates that the “net” effect of GABAergic signaling is inhibitory by the end of the first postnatal week (Murata et al., 2020; Valeeva et al., 2016). This conclusion is supported by in vitro data demonstrating a concurrent increase in chloride extrusion capacity (Spoljaric et al., 2017) and a corresponding decrease in intracellular steady-state chloride concentration (Tyzio et al., 2007; Tyzio et al., 2008). We have cited these papers in the manuscript. Clearly, however, an excitatory action of some GABAergic synapses cannot be entirely excluded at present.

To address the reviewer’s concern, we substantially extended our computational modeling to explore whether alternative network models (or mechanisms) are better suited to explain the observed neuronal dynamics (new Figure S5, new Figure 9). One of these models is a mono-stable network with excitatory GABA, which, however, cannot reproduce important experimental observations. Our data rather indicate that the bi-stable STP-RNN model with inhibitory GABA (as introduced in the initial manuscript version) best explains our experimental data. We added new Figure 9, new Figure S5 and extended the Results section (lines 401–436).

There is a lack of explanation of why P4 networks have more in common with P18 networks than P11 networks, in many cases. The data clearly demonstrates that there is not a progression of network dynamics as a function of age, and instead P11 is, for many measures, behaving differently than both younger and older developmental stages.

While this is true for some of the analyzed parameters, the overall picture that emerged from our analyses is that P4 networks are considerably more similar to networks at P11 than P18 For example, this can be seen in Figure 3A illustrating that activity is exclusively discontinuous at P4, but mostly continuous at P18 – while, at P11, continuous activity occurs for the first time during postnatal development when CA1 dynamically transitions between these two states. We thank the reviewer for his/her critique which prompted us to re-write the first section of Discussion. In this section, we now provide a summary of the major developmental trajectories of network dynamics in CA1 (lines 463–500).

Analysis of FOVs is performed on separate animals at the different ages. The findings would be further strengthened if the same neurons were tracked over time. This can be done in adult mice. However, given the developmental changes that occur between P4 and P18, this type of experiment may not be feasible with current methods. Still, it would be insightful to observe how the same network develops throughout this period. An experiment for the future, perhaps.

We agree with the reviewer that tracking neurons longitudinally would be informative and an interesting experiment for the future. Unfortunately, the methodology has not yet been established for early postnatal mice.

It is not explicitly clear whether the mice are awake during the experiments or under anesthesia. It is stated that head-fixed animals are "spontaneously breathing" in the Results section, and in the Methods they state "Isoflurane was discontinued after completion of the surgical preparation and gradually substituted with the analgesic sedative nitrous oxide". So, what is the general state of the animals during in vivo imaging? This is important as it will certainly affect network activity in CA1 and should be discussed.

We agree that this is an important point of concern. Our rationale for using nitrous oxide (N_2_O) was/is based on our previous studies in the visual cortex of neonatal mice demonstrating that N_2_O has little effect on network activity while profoundly reducing animal movements and, thus, minimizing mechanical artefacts (Kirmse et al., 2015; Kummer et al., 2016).

To further address this important point in the developing CA1, we performed an additional set of experiments in which we compare network activity measured under nitrous oxide with that in unanesthetized mice using a paired experimental design at P11. While our data reveal minor quantitative differences between the two conditions, we demonstrate that major qualitative aspects of the analyzed neuronal dynamics are preserved. Our data further indicate that, unlike conventional anesthetics, N_2_O does not affect respiration or heart rate, while reducing body movements by several-fold. The latter leads to a significant reduction in periods of z-drifts, which cannot be compensated by post-hoc stack alignment and thus need to be excluded from analysis. We added the new Figure 6, new Figure S2 and new Figure S3 and extended the Results (lines 265–286) and Methods (lines 1001–1010 and 1182–1193) sections to illustrate these novel results.

We thank the reviewers for his/her suggestion, as the new dataset clearly augments the relevance and validity of our study.

The data is interpreted that NBs are more prevalent at P11 than P4 and P18. However, given the overall increase in CAT frequency at P18 (Figure 3A, right) it might make it more difficult to detect isolated NBs from those riding on top of (or very close in time to) other NBs (especially given that calcium transients have relatively slow decay kinetics). The authors should be careful to make sure their NB detection method is not biasing them to detect more NBs in FOVs with generally lower activity, i.e. at P11 versus P18.

We thank the reviewer for raising this point of concern. To address this point, we systematically varied our operational definition of NBs in two ways. (1) First, we systematically varied the integration window Δt that is used to compute an activity-dependent threshold for NB detection, separately for each field of view (new Figure S1A). Here, to obtain a statistically justified NB threshold, surrogate data generated by randomly shuffling CaT times were used (see Methods). (2) Second, we used a constant (i.e. activity-independent) threshold which we applied to all fields of view from the three age groups (new Figure S1B). Our results confirm that the reported developmental changes in network burstiness are robust to a wide range of NB definitions. We added the new Figure S1 and extended Results (lines 204–206) and Methods (1104–1110) accordingly.

I do wonder if CAT kinetics differ in PCs at the different age groups. They should show that expression levels are similar, rise times/decay times are similar, noise levels are similar…to rule out these as confounds to the other forms of analysis.

We think that a proper analysis of CaT kinetics cannot be performed, as no electrophysiological ground truth for our Ca^2+^ imaging data is available. As the number of spikes per CaT remains unknown, for this specific purpose, human annotations (as used e.g. in Figure 1) cannot serve as a substitute either. However, we quantified noise levels for all analyzed cells and found them to be similar across the age groups. We added a comment to Methods (lines 1041–1043) and statistical information to Table S2.

Reviewer #3 (Recommendations for the authors):The manuscript addresses an important topic. The data and modeling results provide new insights into the developmental trajectories of network activity in hippocampal CA1 area.

We appreciate the reviewer’s positive feedback.

However, several major aspects, especially concerning (i) the solidity of data from a rather low number of mice, (ii) the lack of experimental data uncovering the underlying mechanisms of described processes and (iii) the interpretation of results in the context of existing literature, dampen my enthusiasm and need to be addressed as part of a major revision before further consideration of the study.

Thank you for the constructive criticism. We comprehensively addressed the reviewer’s points of concerns with additional experiments, analyses and computational modeling (for details, please see below).

1. The in vivo dataset is too small to enable reliable conclusions. In the manuscript, the number of mice used for each analysis is not specified. In general, n numbers should be stated more clearly and be included in the figure legends. For each mouse 3-5 FOVs and in average 14 FOVs/group were acquired, implying that only around 3-4 mice were used for group analysis. Furthermore, the applied stats use FOVs as statistical unit and covers not for single datapoint independency, i.e. FOVs that are coming from the same mouse. The authors might think about the use of mixed-effect models for statistical analysis after increasing the size of the dataset. Given the small size of the recording area, the authors should state how overlapping FOVs and thus, cell populations, between imaging sessions were avoided. Moreover, what was the rationale for focusing the investigation on P4, P11, and P18? Are these time points of particular relevance? In the absence of more time points, it is unclear how the dynamics of described processes evolve.

The dataset of the initial manuscript comprised 19 FOVs from six mice at P4, 11 FOVs from six mice at P11 and 12 FOVs from six mice at P18. Following the editor’s guidance, we recorded and included a novel dataset from another 12 FOVs from six mice at P11 in which the effects of nitrous oxide on animal state and neural dynamics were assessed (see below). In the revised manuscript, we specify the number of mice in Results (line 140–141; line 272) and also in the respective legends of Figure 2 and Figure 6. For each animal, spontaneous activity was recorded from 3–5 FOVs. Some FOVs needed to be excluded from further analysis due to excessive z-drifts. Finally, 1–4 FOVs were analyzed per animal and used for statistics. We extended the corresponding Methods section (lines 1027–1030). As the number of FOVs per mouse is low, a mixed-model ANOVA (or similar) is not applicable in our opinion. Collectively, we are convinced that the included datasets are sufficiently large for detailed quantitative comparisons.

Using two-photon imaging and taking into consideration our restriction to somatic signals, avoiding spatial overlap between sequentially recorded FOVs is straightforward (based on visual control and xyz-coordinates of the objective). We added a sentence to Methods to emphasize that appropriate care was taken to avoid such overlap in our experiments (lines 1029–1030).

We decided to focus on P4–P11–P18 as previous cellular recordings in neocortex demonstrated that, within that time period, the dynamics of developing cortical circuits undergo substantial maturation, which prepares them for e.g. patterned vision and active environmental exploration after eye opening (Kirmse et al., 2022).

2. Besides modeling, additional experimental evidence of the cellular interactions underlying the developmental dynamics of bursts should be added. Direct targeting of distinct neuronal populations, their acute or chronic manipulation, possible combination with electrophysiological recordings, are just few suggestions, how the insights from modeling should be complemented. Moreover, the RNN model provides insights into the mechanisms governing the elevated burstiness constrained to the P11 age group. However, it remains unclear, which mechanisms potentially contribute to the developmental emergence of a bi-stable network as well as its potential disappearance. The authors might include this developmental aspect in the model as well and discuss age-dependent features in synaptic strength and timing that account for observed changes in network synchrony and burstiness in more detail.

Regarding additional experimental investigations, we followed the editor’s guidance by including a novel dataset on the effect of nitrous oxide (see below). We agree that our study raises several interesting follow-up questions, which could be addressed in the future. We feel, however, that analyzing the developmental effects through actual experimental perturbations and/or performing electrophysiological paired recordings in vivo is far beyond the scope of the present manuscript.

To further investigate the mechanisms leading to the development of a bi-stable network, we substantially extended our computational modeling. Our results portend, for instance, that the emergence of a persistent active state in CA1 reflects the developmental strengthening of both GABAergic inhibition and glutamatergic excitation (new Figure 9). We also explore alternative (mono-stable) network models and found that an STP-RNN with inhibitory GABA is best suited to explain the experimental observations (new Figure S5). The impact of input strength and timing, also in relation to the network’s operational state, are analyzed and discussed. We added a new Figure 9, a new Figure S5 and extended the Results section (lines 401–461) accordingly. We thank the reviewer for the specific suggestion.

3. line 371-374: the authors conclude the presence of a lower synchrony in the developing CA1 area compared to sensory cortices as the result of the identification of lower correlation values. The reference cited for visual cortex (Rochefort at al., 2009) uses no pairwise correlation analysis and should be removed. The other references for somatosensory cortices quantify pairwise correlation but use different analytical strategies and not STTC as used in the present manuscript. Thus, the comparison of absolute correlation values might be inappropriate and further depend on the chosen timescale. Another important factor impacting correlation values is the use of anesthesia. Mice were anesthetized with nitrous oxide, known to alter neurotransmission and consequently affecting physiological activity. While anesthesia increases correlation in sensory areas (Goltstein et al., 2015), it decreases STTC values in the CA1 area (Yang et al., 2021). The studies cited in line 371-374 are done in non-anesthetized or in urethane/isoflurane anesthetized mice. Consequently, the identified lower correlation values could thus be an artifact of differential actions of anesthesia in sensory areas versus CA1 area. Moreover, anesthesia has been identified to impact brain activity in an age- and dose-dependent manner (Chini et al., 2019) and might therefore impact the selected age groups differently. Accordingly, the authors should refrain from the statement of a developmental lower correlation in CA1 area than in sensory areas. To support this statement additional recordings in non-anesthetized mice in CA1 area and compared to equally analyzed open-access datasets of sensory cortices would be required. In line with this, PSD analysis of frequencies below 1 Hz as used in Figure 3B are in particular sensitive to anesthesia and should be interpreted with caution.

Correlations: We agree that a quantitative comparison of correlation values is hardly possible and therefore decided to remove this aspect from the first section of Discussion as it is inessential for our main line of argumentation. Instead, we re-wrote the first section of Discussion, where we now provide a summary of the major developmental trajectories of network dynamics in CA1 (lines 463–500).

Anesthesia: We agree that this is an important point of concern. Our rationale for using nitrous oxide (N_2_O) was/is based on our previous studies in the visual cortex of neonatal mice demonstrating that N_2_O has little effect on network activity while profoundly reducing animal movements and, thus, minimizing mechanical artefacts during imaging (Kirmse et al., 2015; Kummer et al., 2016). To further address this important point in the developing CA1, we performed an additional set of experiments in which we compare network activity measured under nitrous oxide with that in unanesthetized mice using a paired experimental design at P11. While our data reveal minor quantitative differences between the two conditions, we demonstrate that major qualitative aspects of the analyzed neuronal dynamics are preserved. Our data further indicate that, unlike conventional anesthetics, N_2_O does not affect respiration or heart rate, while reducing body movements by several-fold. The latter leads to a significant reduction in periods of z-drifts, which cannot be compensated by post-hoc stack alignment and thus need to be excluded from analysis. We added the new Figure 6, new Figure S2 and new Figure S3. We also extended the Results (lines 265–286) and Methods (lines 1001–1010 and 1182– 1193) section to illustrate these novel results. Two of the mentioned papers are cited in Results to further motivate this novel set of experiments (line 270). We thank the reviewers for his/her suggestion, as the new dataset clearly augments the relevance and validity of our study. However, we need to refrain from performing any quantitative comparisons including thirdparty datasets due to differences in recording conditions.

PSD analysis: Please note that the analysis was performed on the fraction of active cells *Φ*(t), which is based on binary time-series data reflecting CaT onset times. Therefore, low-frequency noise in our PSD analyses is not considered problematic. We agree that this is (or could be) different for other data types such as LFP data. Beyond that, we show that nitrous oxide does not affect the PSD of the fraction of active cells *Φ*(t) (Figure 6F and Figure S3D).

4. The threshold for burst detection was quantified individually for each FOV. Consequently, the proportion of silent periods in each FOV affects the threshold calculation and might affect the P11 age group, where activity is almost but not completely continuous, differently. Analysis of bursts detected with a threshold quantified only on "active" periods by excluding silent periods could further support the presence of burst activity during events without be affected by the changing discontinuity over age.

We thank the reviewer for raising this point of concern. We are aware of the fact that any definition of network bursts is operational in nature. As correctly pointed out by the reviewer, NB threshold in our definition depends on the amount of silent periods, but this is actually intended. One technical reason for not restricting our analyses to “active” periods is that such an approach would require a number of additional assumptions (e.g. related to the definition “active” vs. “inactive” periods), which render the descriptive value of the obtained results less intuitive. Moreover, in biological terms, discarding “inactive” periods would artificially reduce the signal-to-noise ratio of NBs during periods of discontinuous activity and thus introduce (another type of) bias.

To address the robustness of the reported developmental changes in NBs, we systematically varied our operational definition of NBs in two ways. (1) First, we systematically varied the integration window Δt that is used to compute an activity-dependent threshold for NB detection, separately for each field of view (new Figure S1A). (2) Second, we used a constant (i.e. activityindependent) threshold which we applied to all fields of view from the three age groups (new Figure S1B). Our results confirm that the reported developmental changes in network burstiness are robust to a wide range of NB definitions (lines 204–206).

5. The authors describe in the Introduction that in the neonatal hippocampus activity is triggered by myoclonic twitches. Did the authors monitor the animal's movement? If yes, the occurrence of bursts and presence of network motifs could be correlated to the movement. This would enable a better understanding of how age-dependent dissociation of CA1 activity from twitches relates to the emergence and stabilization of self-organized hippocampal motifs during recurring activation patterns (line 258, 259) and changes in neuronal synchrony.

Using wide-field Ca^2+^ imaging, we recently found that only about half of sharp waves (SPWs) are movement-related (Graf et al., 2021). In addition, the same study revealed a second class of SPW- and movement-independent events that account for approximately two thirds of the total activity (Graf et al., 2021). A previous electrophysiological study showed that SPWs increasingly decouple from myoclonic twitches throughout the first postnatal week so that by ~P9, spontaneous movements and SPWs rarely co-occur (Mohns et al., 2007). More recently, this decoupling has been confirmed at the single-cell level (Dard et al., 2022). Additionally, (brief) myoclonic twitches are rare by the third postnatal week, while longer movement periods increase over development, indicating that the characteristics of body movements undergo substantial developmental changes too. We therefore feel that an extensive analysis of how different forms of movements (twitches, startles, more complex motor behaviors) influence neuronal dynamics at different developmental stages is beyond the scope of our manuscript. We further have some technical concerns: First, (vigorous) movements tend to cause z-drifts in two-photon imaging, which cannot be compensated through stack registration methods and thus need to be excluded from analysis. Second, we monitored animal movement using a single pressure sensor positioned below the chest of the mouse, which probably is insufficient for differentiating these different types of motor behaviors throughout development. Third, the recorded pressure signal was intended to be used for monitoring anesthesia and animal state during the experiment, but is unfortunately not perfectly synchronized to the scanning mirror movements (+/- 1 to 2 s, with the exact delay being unknown), which effectively prevents a reliable analysis.

[Editors’ note: what follows is the authors’ response to the second round of review.]

The manuscript has been improved but there are some remaining issues that need to be addressed, as outlined below:While all the reviewers remain enthusiastic about this work, Reviewer 3 points out some major concerns that need to be addressed before this manuscript can be recommended for publication.

We are pleased about the reviewers' enthusiasm for our manuscript.

We provide a detailed point-to-point reply to the concerns listed below in the “Reviewer #3 (Recommendations for the authors)” section.

We further adapted the formatting of our manuscript as detailed by the *eLife* editorial support.

Reviewer #3 (Recommendations for the authors):The manuscript by Graf et al., investigates the population dynamics of the mouse CA1 hippocampal area across the first two weeks of extra-uterine life. The manuscript leverages the combination of experimental data and modeling to identify P11 as a developmental stage in which network burstiness peaks due to network bi-stability. An increase in synaptic inhibition is suggested as being the reason behind this transient network characteristic.

We very much appreciate the reviewer’s detailed evaluation of our study.

The revised manuscript employs a series of innovative and state-of-the-art analytical techniques and the modeling work is elegantly used to try to get mechanistic insight into the processes underlying the network properties and how they evolve throughout development. The modeling section of the paper is very high quality, yet it is not always clear how it relates (or how it explains) the experimental data that is presented in the first part of the manuscript (see below for comments). The authors try to explain the link between experimental and modelling work by discussing published data from in vitro, electrophysiological and imaging studies in CA1 but also in sensory areas. Due to the methodological variability as well as diverse selected age ranges in these studies, it is unclear how they relate to the ones included in the manuscript. Although the authors might refrain from doing manipulation experiments, they could image GABAergic neurons to better understand their contributions to NBs.

We fully agree that detailed information on the participation of GABAergic interneurons would be highly valuable, especially as it would allow us to further constrain computational models. Simultaneous imaging from molecularly distinct populations using genetically encoded Ca^2+^ indicators is, however, far from trivial, particularly at early developmental stages. While we think that addressing this point is experimentally very challenging and beyond the scope of the present manuscript, we added this important consideration to Discussion (l. 567–569), as it might guide future research in the field.

In line with the concerns listed below, the manuscript does provide solid experimental and theoretical evidence for its conclusions.

We appreciate that the reviewer considers our conclusions to be based on “solid experimental and theoretical evidence”. We also thank the reviewer for her/his valuable suggestions on how to further improve the manuscript. Please find our detailed point-by-point reply below.

I appreciate the work that was done to investigate the influence that N20 anesthesia has on hippocampal network activity, but I find that the conclusions that the authors draw from these experiments is not rooted in the data. Furthermore, and perhaps more concerning, the network activity of P11 "anesthetized" mice in this dataset seemingly contradict the main results that are presented in the remainder of the manuscript.– The activity of anesthetized P11 mice displayed in figure 6 greatly differs from anesthetized age-matched mice in the rest of the manuscript.

Please note that the paired experiments shown in Figure 6 required substantially longer recording times as compared to our previous measurements. To prevent photo-bleaching and photo-toxicity in these paired experiments, it was therefore necessary (i) to reduce the laser power and (ii) to increase the detector (PMT) gain. This in turn effectively decreased the signalto-noise ratio in the dataset of Figure 6 (N_2_O vs. unanesthetized at P11) and is the likely reason for the lower CaT frequencies in this vs. the previous (P4–11–18 under N_2_O) dataset. We had mentioned this methodological consideration in the Methods section of the previous manuscript but agree that it should be made more explicit to the reader already in Results. We therefore added a sentence to l. 289–291.

As a result of this systematic difference in recording conditions, quantitative comparisons between the previous and novel dataset will be biased. Therefore, in Results, we needed to refrain from performing statistical analyses on absolute values. In qualitative terms, network activity measured in the novel dataset resembles that of the previous one in several important aspects such as burstiness and continuity and, thus, strengthens our results presented in the remainder of the manuscript.

– The average theta bandpower of N20 mice in figure S3D is less than half of those of age-matched mice in figure 3E (~3.7x10-4 vs 8.9x10-4). These values would be much more similar to P4 (2.9x10-4) and P18 (3.4x10-4) mice than to the P11 ones (8.9x10-4) that are plotted in figure 3E.

This is most likely a direct consequence of the systematic difference in recording conditions detailed above, as lower mean frequencies of CaTs are expected to result in a lower absolute power. We would like to emphasize that the shape of the PSD with a prominent peak at ~0.1-0.5 Hz (i) was similar between the previous (Figure 3) and novel (Figure 6) datasets at P11 and (ii) was unaffected by N_2_O (Figure 6). We would also like to note that, for clarity, we aim at avoiding terms such as “theta”, as they were defined in the context of EEG/LFP data, whereas here we subjected the fraction of active cells *Φ*(t) (based on detected CaTs) to a power spectrum analysis to quantify the recurrence of neuronal coactivation.

– The peak in the Φ PSD of figure 6F for both anesthetized and unanesthetized mice is virtually indistinguishable to that of P18 mice in figure 3D (and roughly half the size of that of P11 mice).

For the reasons detailed above, any quantitative statistical comparisons between the previous (Figure 3) and novel (Figure 6) datasets will be biased by the difference in recording conditions. Therefore, in Results, we needed to refrain from performing statistical analyses on absolute values.

– The N20 mice in figure S3G have a % of cells with a significant STTC that is less than half than age-matched mice in figure 4E. The values in figure S3G are actually much closer to the P18 group than the P4 one. How can that be?

Yes, this is also a direct consequence of the systematic difference in recording conditions detailed above. Lower signal-to-ratio (see above) is expected to decrease the detected number of active neurons (e. g. during network bursts), which can in turn result in lower STTC values.

– Could the authors verify whether anesthesia affects other parameters that are central to the thesis of the manuscript such as the Gini coefficient of CaT etc.?

We thank the reviewer for this suggestion. We added (i) the Gini coefficients of CaT frequencies to the new Figure 6—figure supplement 2D (l. 293–294) and (ii) the time spent in continuous activity to the new Figure 6—figure supplement 2E (l. 294–295). The complete statistical information was added to Supplementary file 1f.

– I find the provided data does not corroborate the statement that neuronal network activity under nitrous oxide closely resembles that recorded in unanesthetized mice. Several parameters that are important to the thesis of the manuscript such as STTC, population coupling etc. are affected by anesthesia.

We followed the reviewer’s suggestion and rephrased our conclusion (l. 302–305).

Network burstiness was not significantly affected by N_2_O (l. 294–297 in Results and also added to Discussion in l. 528–530). This is indeed an important finding which augments the relevance and validity of our computational modeling.

As a general concern, providing a comparison between anesthetized and unanesthetized mice at one single developmental timepoint does not give much insight into whether the developmental processes that are described in the paper are biased by anesthesia. While experimentally addressing this concern is time consuming, it should be discussed.

We appreciate the reviewer’s suggestion and extended the Discussion section accordingly (l. 530–534).

Line 125: what is this event detection routine based on analyzing mean ∆F(t) that you compared CATHARSIS to?

We thank the reviewer for identifying this ambiguity. In the ‘mean ∆F(t) approach’, for a given ROI, we first computed ΔF(t) by frame-wise averaging over all pixels belonging to that ROI. We then extracted CaT onsets from ΔF(t) using UFARSA, a general-purpose event detection routine (Rahmati et al., 2018). We added this information to Methods (l. 1207–1210).

Overall, the information provided to assess the quality of the calcium transient extraction pipeline is scarce, and its validity has to be taken at face value. It would be comparable to an electrophysiology paper using its unique spike sorting algorithm. While I can understand that extracting calcium transients from a densely packed brain area such as the rodent CA1 has its own unique challenges, this is now routinely done using established pipelines. For instance, a recent paper (Dard et al., 2022, also published in eLife) used suite2p to extract calcium transients from the developing CA1 network (same developmental phase). I would suggest the authors to attempt at replicating their results using this more established calcium transient pipeline.

We thank the reviewer for this helpful comment. In the cited paper (Dard et al., 2022), Suite2P was used for cell segmentation only, whereas event detection was performed using DeepCINAC, a novel algorithm developed by the same laboratory (Denis et al., 2020). DeepCINAC and CATHARSiS were developed for the same reason: Both the Cossart and our laboratories concluded that previously published pipelines are less suited to deal with high false-positive rates arising from “highly synchronous neurons located in densely packed regions such as the CA1 pyramidal layer of the hippocampus during early postnatal stages of development” (Denis et al., 2020). While the algorithmic approaches differ, both CATHARSiS and DeepCINAC make use of the full spatial ΔF profile. In terms of performance, CATHARSiS offers better detection results than a conventional ‘mean ∆F(t) approach’, and DeepCINAC was found to offer better detection results than CaImAn, another widely used pipeline (Giovannucci et al., 2019). In the absence of ground-truth datasets, both CATHARSiS and DeepCINAC were evaluated based on human consensus annotations. We therefore doubt that more popular pipelines, which still require extensive parameterization, provide bona fide superior detection results. Moreover, Suite2P uses tailored methods (subtraction) to remove neuropil/background contamination which can also lead to false removal/attenuation of veridical CaTs in densely packed regions displaying high synchrony. Indeed, one of the main reasons to develop CATHARSIS was to carefully demix the time-series of each ROI from its overlapping ROIs and neuropil activity. This is the reason why we evaluated our new approach based on both simulated (Figures 1A–D) and measured (Figures 1E–I) data, where the measured data were acquired under the recording conditions used in the manuscript. We hope that we clarified that our method is, currently, difficult to compare to some gold standard but we have taken care to verify our results. We added a sentence to Results (l. 99–102) to make the reader aware of recent papers demonstrating that popular CaT analysis algorithms can produce substantial misattribution errors (false positives) under similar conditions.

Line 189: figure 3C and the manner in which the term "continuous activity" is used in this paragraph is perplexing. Brain activity is discontinuous (alternation of low-activity (silent) state and high activity periods) in early development, but it should already be continuous and adult-like at P18 if not already at P11. It is therefore perplexing that several P18 mice have a time in continuous activity (Figure 2C) that is well below 50% and a few even below 20%. If this is an effect of anesthesia, it is concerning and a datapoint that goes in the direction of N2O having a major impact on hippocampal activity (anesthesia seems indeed to reduce CaT frequency also in the data presented here, see Figure S3C). If this depends on the manner in which "continuous activity" is defined/computed, perhaps a different term should be used.

Yes, the reviewer is correct. We employ an operational definition of continuous activity that is based on somatic CaTs (a proxy of firing) recorded from small local networks in the order of ~100–200 CA1 neurons. Our definition is based on the fraction of active cells over time and thus inevitably differs from definitions based on e.g. LFP data. To make the reader aware of this important point, we rephrased the Results section (l. 189–196) and added a comment to Methods (l. 1245 and l. 1250–1251).

Since N_2_O did not affect continuity defined in this manner (new Figure 6—figure supplement 2E and revised Supplementary file 1f), the obtained values (at P11) are not an artefact of anesthesia.

The authors write that a statistical mixed-effect model is not applicable in their opinion due to low numbers of FOVs/mouse. However, the data are not independent and a mouse with 4 FOVs biases the data distributions much stronger than a mouse with only 1 FOV. This is especially important since an overlap of FOVs cannot be ruled out as they write in their methods. If the authors don't want to use mixed-effect models, they should take mice as statistical unit by taking the average of each mouse, as it's also done in many of their cited studies. A minor point here, in Figure 4 D the number of neurons is written in the legends. For P11 mice 11 FOVs and for P18 mice 12 FOVs were recorded, resulting in 161 neurons for P11 and 100 neurons for P18, respectively. How do the authors explain the substantially higher yield in P11 mice? Might differential effects of anesthesia play a role?

We apologize for being unclear regarding a potential overlap of FOVs. In fact, any spatial overlap between sequentially recorded FOVs was strictly avoided based on the xyzcoordinates of the objective and visual control. We rephrased the respective sentence in Methods for clarity (l. 1150–1151). That is, each analyzed cell contributed to exactly one FOV only, implying that the minimum requirement for using the number of FOVs as the statistical parameter n is met.

However, we fully understand the reviewer’s concern and agree that mice contributing a larger number of FOVs have a greater statistical weight in our analyses. Conversely, post-hoc averaging across FOVs is also problematic for at least two reasons: (i) The lower the number of FOVs per mouse, the greater the weight of a single FOV. Hence, post-hoc averaging across FOVs would introduce another kind of bias that is opposite in direction to the one pointed out by the reviewer. (ii) More fundamentally, several of our analyses (e.g., pairwise correlations, population coupling, NBs, and motifs) are based on the simultaneous sampling of activity from multiple cells. This effectively renders the FOV (not the animal) our analytical unit. We therefore decided not to modify our analytical approach. However, we (i) added our reasoning to the “Methods – Statistical analysis” section (l. 1451–1457) and, to complement the descriptive statistics, (ii) also provide the mouse and FOV IDs in the Figure 2-souce data 1.

Regarding Figure 4D, the given numbers of neurons represent individual FOVs. For example, in the P4 example on the left of Figure 4D, 124 neurons were analyzed for the depicted representative FOV. We added a comment to the legend of Figure 4 to clarify this point (l. 938).

I appreciate that, in the revised manuscript, the authors considered the concerns regarding the unwanted effects of anesthesia and performed additional experiments in unanesthetized P11 mice. However, as they write, anesthesia has age-dependent effects on network activity. Especially, the emergence of an active sleep-wake cycle (at ~P14) is suggested to co-occur with frequency-specific (i.e. altered network dynamics) effects of anesthesia (Ackman et al., 2014; Chini et al., 2019; Cirelli and Tononi, 2015; Shen and Colonnese, 2016). Thus, their anesthetic strategy might affect their comparisons between P18 and younger mice.

We thank the reviewer for raising this point. We extended the Discussion accordingly (l. 530– 534). The mentioned papers were cited in l. 280–281 and/or l. 533–534.